# Adaptive numerical solution of Kadanoff-Baym equations

**Francisco Meirinhos**[1*], **Michael Kajan**[1], **Johann Kroha**[1] **and Tim Bode**[2†]

**1** Physikalisches Institut and Bethe Center for Theoretical Physics,
Universität Bonn, Nussallee 12, 53115 Bonn, Germany
**2** German Aerospace Center (DLR), Linder Höhe, 51147 Cologne, Germany

⋆ meirinhos@physik.uni-bonn.de , † tim.bode@dlr.de

## Abstract

A time-stepping scheme with adaptivity in both the step size and the integration order is presented in the context of non-equilibrium dynamics described via Kadanoff-Baym equations. The accuracy and effectiveness of the algorithm are analysed by obtaining numerical solutions of exactly solvable models. We find a significant reduction in the number of time-steps compared to fixed-step methods. Due to the at least quadratic scaling of Kadanoff-Baym equations, reducing the amount of steps can dramatically increase the accessible integration time, opening the door for the study of long-time dynamics in interacting systems. A selection of illustrative examples is provided, among them interacting and open quantum systems as well as classical stochastic processes. An open-source implementation of our algorithm in the scientific-computing language Julia is made available.

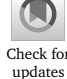
# 1  Introduction

The research fields requiring the solution of time-dependent, non-equilibrium many-body problems comprise, among others, cosmology and high-energy particle physics [1–3], spin dynamics [4], cold atoms [5–8], superconductivity [9], the Kondo effect [10, 11] and other strongly correlated electronic systems [12–17]. Due to the great difficulty in addressing such problems with purely analytical methods, their numerical solution is an active area of research [18–21] and plays a central role in the understanding of time-dependent many-body phenomena.

Numerical methods such as exact diagonalisation [22], the time-dependent density-matrix renormalisation group [23] or real-time quantum Monte Carlo (QMC) [24] have lead to great insight into the time evolution of many-body systems and can account for arbitrarily strong interactions. However, they are limited in system size, particle number and evolution time. Real-time QMC, for instance, may suffer from dynamical sign problems, limiting the solution to the short-time dynamics [25]. Matrix-product-state methods as well as exact diagonalisation are limited by entanglement growth and generally scale exponentially with the system size.

By contrast, quantum field-theoretical formulations of many-body problems are in general approximate but can incorporate large particle numbers. The framework of non-equilibrium quantum field theory (NEQFT) is the Schwinger-Keldysh formalism [3, 26–29]. Although it requires sophisticated techniques such as $1/\mathcal{N}$ expansions [3] or pseudoparticle formulations [30] whenever perturbation theory fails, it generally scales more favourably with system size and evolution time. Thus, it can offer significant physical insight into systems and regimes difficult to address otherwise.

Another important area of interest to which the NEQFT framework can be straightforwardly applied are open quantum systems. These are frequently treated by the Lindblad master equation [31, 32], while the entirely equivalent approach via the Schwinger-Keldysh formalism [29, 33, 34], even though less widespread, opens up the large toolbox of NEQFT for these applications. Already for non-interacting, open systems, NEQFT can be quite adequate when computing two-time expectation values, while obtaining the latter via direct numerics quickly becomes prohibitive.

Last but not least, the Schwinger-Keldysh formalism demonstrates the close relationship

between the formalisms of quantum and classical statistical physics [28,34,35] by bearing out explicitly the connection to the Martin-Siggia-Rose (MSR) formalism [36] of classical statistical field theory, the path-integral formulation of which was elaborated by Janssen [37] and De Dominicis [38]. Classical stochastic processes are important in a great number of fields such as in, e.g., quantum optics [39,40], active matter [41,42], chemistry [43,44], financial markets [45]. Via the MSR formalism, it is possible to subsume classical stochastic processes [43, 44] under the methods employed in this paper, thus providing an alternative tool beyond the ubiquitous stochastic differential and Fokker-Planck equations. Since the Schwinger-Keldysh formalism directly provides the Green function, an expectation value which in a statistical sense essentially represents the second cumulants of the system, and which is usually sufficient to construct the physical quantities of interest, the NE(Q)FT approach can be preferable. The field-theoretical derivation of the dynamical equations for the cumulants is not only elegant but also allows for systematic and controlled (perturbative) approximations in the presence of non-linearities [46].

Common to all of the above applications is that their full solution requires the computation of expectation values that, on the level of the second cumulants, depend on *two* times. Such two-point functions are fundamental objects of many-body physics as they describe, for instance, single-particle excitations and statistical particle distributions, which represent the essential part of the experimentally accessible observables. Computing these time-dependent correlation functions thus appears to be a universal problem whose solution demands general and efficient numerical tools. The dynamical equations arising from the NE(Q)FT of all of these problems are generally known as *Kadanoff-Baym* (KB) equations [47], a set of two-time non-linear integro-differential equations. The distinct non-Markovian structure of the KB (integro-differential) equations arises from the reduction of the state space from the (differential) equations generating the Martin-Schwinger hierarchy – with Markovian structure but dependence on *all n*-point functions. Similar to the computation of Martin-Schwinger hierarchy, an exact computation of the KB equations is an intractable problem, for which truncations of the interaction diagrams will be required. Furthermore, the computational effort to solve the KB equations has led to a number of approximation techniques, among them memory truncation [18,48], the generalised Kadanoff-Baym ansatz [20,49], as well as advanced computational methods such as high-order time-stepping algorithms [19], parallelised programming [50], finite-element representations [20,51], and data compression [18].

Lacking so far in the numerical integration of the KB equations is a technique that is common in the solution of standard ordinary differential equations: *adaptivity*. Due to the considerable computational cost of solving the KB equations – with operations scaling at least as $\mathcal{O}(n^2)$, where $n$ is the number of time-steps – it is desirable to minimise this number. Moreover, the time evolution of non-equilibrium problems may contain several different timescales at different times [14], which can suitably be captured with adaptive schemes.

In this paper, we present an integration algorithm for the KB equations which is adaptive both in the step size and in the integration order. We substantiate the effectiveness of our algorithm by obtaining numerical solutions of exactly solvable models (both quantum and classical) as well as non-trivial interacting quantum systems. This paper is organised as follows. In Section 2 we outline the field theory required to derive the KB equation for both classical and quantum systems. In Section 3 the adaptive scheme for the numerical solution of KB equations is presented. In Section 4 a series of classical and quantum problems are solved to benchmark and showcase the adaptive scheme. We summarise our work in Section 5.

# 2 Non-Equilibrium Field Theory

In this section, we succinctly introduce the field-theory framework used to derive the Kadanoff-Baym equations to be solved numerically later on. We begin with an introduction to the so-called *two-particle irreducible* (2PI) effective action [2,3,52] as a versatile and rigorous way to construct self-consistent and conserving [47] Dyson equations for the non-equilibrium Green functions [53]. The conserving properties can also be understood from the perspective of "Φ-derivable" approximations [54], for the construction of which the 2PI effective action provides a well-defined pathway. In the second part of this section, 2.2, we give a brief discussion of the Schwinger-Keldysh formalism for both closed and open quantum systems. From the latter, it will then be possible to include classical stochastic systems into the framework. Note that for the sake of simplicity, we focus solely on non-correlated (i.e. Gaussian) initial states. Hence we omit the "vertical" branch [55] typically attached to the Schwinger-Keldysh contour (Fig. 1) for correlated equilibrium initial states. Throughout this work, we set $\hbar = 1$ .

## 2.1 Two-Particle Irreducible Effective Action

For a system described by an action functional $S[\boldsymbol{\phi}(t)]$, where $\boldsymbol{\phi}(t) = (\phi_1(t), \ldots, \phi_N(t))^T$ is a vector of fields (real scalar and/or real Grassmann) that contains all relevant degrees of freedom (space, spin, different field components, etc), the moment-generating or partition function is defined as

$$Z[\boldsymbol{j}, \boldsymbol{K}] = \int \mathcal{D}\boldsymbol{\phi} \exp\left\{ i\left[ S[\boldsymbol{\phi}(t)] + \int dt\, \boldsymbol{j}^T(t)\boldsymbol{\phi}(t) + \frac{1}{2} \int dt dt'\, \boldsymbol{\phi}^T(t) \boldsymbol{K}(t,t') \boldsymbol{\phi}(t') \right] \right\}, \quad (1)$$

where $\boldsymbol{j}$ and $\boldsymbol{K}$ are 1- and 2-time source fields and $\int \mathcal{D}\boldsymbol{\phi}$ denotes functional integration over all field components $\phi_a(t)$, $a = 1, \ldots, N$. In this notation, complex fields are covered by treating real and imaginary parts as separate components [2, 5]. This gives rise to the so-called cumulant-generating function (CGF)

$$W[\boldsymbol{j}, \boldsymbol{K}] = -i \ln Z[\boldsymbol{j}, \boldsymbol{K}]. \quad (2)$$

For the standard construction of $Z = e^{iW}$ from time-ordered operators by means of coherent states, see Refs. [2, 28, 29, 56], which include discussions of Gaussian initial states and the functional measure. The 2PI effective action $\Gamma$ is now defined as the double Legendre transform of $W$ with respect to the source fields $\boldsymbol{j}$ and $\boldsymbol{K}$,

$$\Gamma[\bar{\boldsymbol{\phi}}, \boldsymbol{G}] = W[\boldsymbol{j}, \boldsymbol{K}] - \int dt\, \boldsymbol{j}^T(t)\bar{\boldsymbol{\phi}}(t) - \frac{1}{2} \int dt dt'\, \text{Tr}\, \boldsymbol{K}(t,t') \left( i\boldsymbol{G}(t,t') + \bar{\boldsymbol{\phi}}(t)\bar{\boldsymbol{\phi}}^T(t') \right),$$

where $\bar{\boldsymbol{\phi}}(t)$ and $i\boldsymbol{G}(t,t')$ are the first and second cumulant, respectively. Their definitions, in components, are given by

$$\begin{aligned} \bar{\phi}_a(t) &= \frac{\delta W}{\delta j_a(t)} = \langle \phi_a(t) \rangle_{j,K}, \\ G_{ab}(t,t') &= -\frac{\delta^2 W}{\delta j_a(t)\delta j_b(t')} = -i\left[ \langle \phi_a(t)\phi_b(t') \rangle - \langle \phi_a(t) \rangle \langle \phi_b(t') \rangle \right]_{j,K}. \end{aligned} \quad (3)$$

In the quantum case, the components of the field vector $\boldsymbol{\phi}(t)$ are understood as living on the Schwinger-Keldysh time contour, which we discuss in the next section (2.2). For completeness, note that the second moment can also be obtained using

$$\frac{\delta W}{\delta K_{ab}(t,t')/2} = \langle \phi_a(t)\phi_b(t') \rangle_{j,K}. \quad (4)$$

In close connection to the one-loop, one-particle irreducible effective action [57], for real scalar fields the 2PI effective action is given by [2, 3]

$$\Gamma[\bar{\boldsymbol{\phi}}, \boldsymbol{G}] = \text{const.} + S[\bar{\boldsymbol{\phi}}] + \frac{\mathrm{i}}{2} \text{Tr} \ln \boldsymbol{G}^{-1} + \frac{\mathrm{i}}{2} \text{Tr} \, \boldsymbol{G}_0^{-1}[\bar{\boldsymbol{\phi}}] \boldsymbol{G} + \Gamma_2[\bar{\boldsymbol{\phi}}, \boldsymbol{G}], \tag{5}$$

where $\Gamma_2[\bar{\boldsymbol{\phi}}, \boldsymbol{G}]$ contains only two-particle irreducible diagrams [2]. A specific example for a bosonic $\Gamma_2$ is given in Eq. (53). Real Grassmann fields, in turn, lead to a 2PI effective action that reads

$$\Gamma[\boldsymbol{G}] = \text{const.} - \frac{\mathrm{i}}{2} \text{Tr} \ln \boldsymbol{G}^{-1} - \frac{\mathrm{i}}{2} \text{Tr} \, \boldsymbol{G}_0^{-1} \boldsymbol{G} + \Gamma_2[\boldsymbol{G}]. \tag{6}$$

Note that an example for a fermionic $\Gamma_2$ functional can be found in Eq. (58). After setting the external sources $\boldsymbol{j}$ and $\boldsymbol{K}$ to zero, $\bar{\boldsymbol{\phi}}$ and $\boldsymbol{G}$ are determined self-consistently via the equations

$$0 = \frac{\delta \Gamma[\bar{\boldsymbol{\phi}}, \boldsymbol{G}]}{\delta \bar{\phi}_a(t)}, \qquad 0 = \frac{\delta \Gamma[\bar{\boldsymbol{\phi}}, \boldsymbol{G}]}{\delta G_{ba}(t', t)}, \tag{7}$$

the first of which determines the equations of motion of the "mean fields" $\bar{\phi}_a(t)$ (if present), while the second one may be written as

$$G_{ab}^{-1}(t, t') = G_{0,ab}^{-1}(t, t') - \Sigma_{ab}(t, t'), \tag{8}$$

which we recognise as Dyson's equation and where the *self-energy* is defined as

$$\Sigma_{ab}(t, t') = \pm 2\mathrm{i} \frac{\delta \Gamma_2}{\delta G_{ba}(t', t)}, \tag{9}$$

for real fields and for real Grassmann fields, respectively. These equations can be brought to KB form and then are given by

$$\boldsymbol{G}_0^{-1}(t)\boldsymbol{G}(t, t') = \delta(t - t')\mathbb{1} + \int \mathrm{d}s \, \boldsymbol{\Sigma}(t, s)\boldsymbol{G}(s, t'). \tag{10}$$

## 2.2 Schwinger-Keldysh Formalism

To describe the time evolution of quantum systems out of equilibrium, the time integrals have to be evaluated over the Schwinger-Keldysh contour $\mathcal{C} = \mathcal{C}^+ \cup \mathcal{C}^-$, which consists of forward (+) and backward (−) branches, as depicted in Fig. 1. This results in the well-known "doubling" of degrees of freedom [28], i.e. the field components are doubled according to

$$\phi_a(t) \longrightarrow \phi_{a,\sigma}(t), \qquad \sigma = \begin{cases} +, \, t \in \mathcal{C}^+, \\ -, \, t \in \mathcal{C}^-, \end{cases} \tag{11}$$

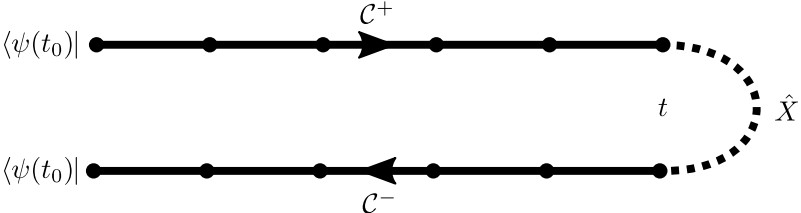

Figure 1: The Schwinger-Keldysh contour arises naturally when calculating time-dependent expectation values:
$\langle \hat{X}(t) \rangle = \langle \psi(t) | \hat{X} | \psi(t) \rangle = \langle \psi(t_0) | \hat{U}(t_0, t) \hat{X} \, \hat{U}(t, t_0) | \psi(t_0) \rangle.$

such that the new field vector becomes, $(\phi_{1,+}(t), \ldots, \phi_{N,+}(t), \phi_{1,-}(t), \ldots, \phi_{N,-}(t))^T$. Exemplary action functionals of this field vector can be found in Eqs. (18) and (67). The corresponding Green functions are then given by

$$
\begin{aligned}
G_{ab}^{\sigma\sigma'}(t,t') &= -\mathrm{i}\left[\langle\phi_{a,\sigma}(t)\phi_{b,\sigma'}(t')\rangle - \langle\phi_{a,\sigma}(t)\rangle\langle\phi_{b,\sigma'}(t')\rangle\right] \\
&= \begin{cases}
\Theta\left(t-t'\right)G_{ab}^{>}(t,t') + \Theta\left(t'-t\right)G_{ab}^{<}(t,t'), & t,t'\in\mathcal{C}^{+}, \\
\Theta\left(t-t'\right)G_{ab}^{<}(t,t') + \Theta\left(t'-t\right)G_{ab}^{>}(t,t'), & t,t'\in\mathcal{C}^{-}, \\
G_{ab}^{<}(t,t'), & t\in\mathcal{C}^{+}, t'\in\mathcal{C}^{-}, \\
G_{ab}^{>}(t,t'), & t\in\mathcal{C}^{-}, t'\in\mathcal{C}^{+},
\end{cases}
\end{aligned}
\tag{12}
$$

where $G^{>}$ and $G^{<}$ the greater and lesser Green functions, respectively. Note that for symmetry-broken bosonic systems, there holds $\langle\phi_{a,+}(t)\rangle = \langle\phi_{a,-}(t)\rangle \neq 0$, whereas $\langle\phi_{a,\pm}(t)\rangle = 0$ in the fermionic case.

### 2.2.1 Closed quantum systems

For closed quantum systems, in practice it is usually more straightforward to develop the self-energy diagrammatically in terms of the time-ordered Green function, using the Hamiltonian in operator form and Wick's theorem. Then only after this step does one have to evaluate the integral of Eq. (10) over the Schwinger-Keldysh contour to obtain explicit equations of motion in terms of $G^{\lessgtr}(t,t')$. This is achieved by writing Eq. (10) as

$$
(\mathrm{i}\partial_t - \boldsymbol{h}_0)\boldsymbol{G}(t,t') = \delta_{\mathcal{C}}(t-t')\mathbb{1} + \int_{\mathcal{C}} \mathrm{d}\bar{t}\,\boldsymbol{\Sigma}(t,\bar{t})\boldsymbol{G}(\bar{t},t'),
\tag{13}
$$

where the matrix $\boldsymbol{h}_0$ denotes single-particle contributions to the Hamiltonian, and the contour integration can be decomposed as

$$
\int_{\mathcal{C}} \mathrm{d}\bar{t} = \int_{t_0}^{\min(t,t')} \mathrm{d}\bar{t} + \int_{\min(t,t')}^{\max(t,t')} \mathrm{d}\bar{t} + \int_{\max(t,t')}^{t_0} \mathrm{d}\bar{t}.
\tag{14}
$$

A sketch of this decomposition given in Fig. 2. For the (contour-ordered) Green function, one then uses the expression

$$
\boldsymbol{G}(t,t') = \Theta_{\mathcal{C}}\left(t-t'\right)\boldsymbol{G}^{>}\left(t,t'\right) + \Theta_{\mathcal{C}}\left(t'-t\right)\boldsymbol{G}^{<}\left(t,t'\right),
\tag{15}
$$

with $\Theta_{\mathcal{C}}\left(t'-t\right)$ the Heaviside step function along the Schwinger-Keldysh contour, and obtains the non-equilibrium equations of motion

$$
\begin{aligned}
(\mathrm{i}\partial_t - \boldsymbol{h}_0)\boldsymbol{G}^{<}(t,t') &= \int_{t_0}^{t} \mathrm{d}\bar{t}\,\boldsymbol{\Sigma}^{>}(t,\bar{t})\boldsymbol{G}^{<}(\bar{t},t') + \int_{t}^{t'} \mathrm{d}\bar{t}\,\boldsymbol{\Sigma}^{<}(t,\bar{t})\boldsymbol{G}^{<}(\bar{t},t') \\
&\quad + \int_{t'}^{t_0} \mathrm{d}\bar{t}\,\boldsymbol{\Sigma}^{<}(t,\bar{t})\boldsymbol{G}^{>}(\bar{t},t'),
\end{aligned}
\tag{16a}
$$

$$
\begin{aligned}
(\mathrm{i}\partial_t - \boldsymbol{h}_0)\boldsymbol{G}^{>}(t,t') &= \int_{t_0}^{t'} \mathrm{d}\bar{t}\,\boldsymbol{\Sigma}^{>}(t,\bar{t})\boldsymbol{G}^{<}(\bar{t},t') + \int_{t'}^{t} \mathrm{d}\bar{t}\,\boldsymbol{\Sigma}^{>}(t,\bar{t})\boldsymbol{G}^{>}(\bar{t},t') \\
&\quad + \int_{t}^{t_0} \mathrm{d}\bar{t}\,\boldsymbol{\Sigma}^{<}(t,\bar{t})\boldsymbol{G}^{>}(\bar{t},t').
\end{aligned}
\tag{16b}
$$



Figure 2: Expanding the contour-ordered Green function $\boldsymbol{G}(t, t')$.

### 2.2.2 Open systems, quantum and classical

Systems in particle exchange with a Markovian reservoir can be described in Born-Markov approximation (second-order perturbation in the bath coupling, bath relaxation time fast compared to the intrinsic system time scales) by a Lindblad form of the master equation for the density matrix $\hat{\rho}$. For illustrative simplicity, here we assume a single-mode cavity with Hamiltonian $\hat{H} = \omega_0 a^\dagger a$ at some low temperature $\beta^{-1} \ll \hbar\omega_0$. Such a system is described by the master equation

$$\partial_t \hat{\rho} = -\mathrm{i}\left[\hat{H}\hat{\rho} - \hat{\rho}\hat{H}^\dagger\right] + \lambda\left(\hat{a}\hat{\rho}\hat{a}^\dagger - \frac{1}{2}\{\hat{a}^\dagger\hat{a}, \hat{\rho}\}\right), \tag{17}$$

where $\lambda$ is the effective loss rate, and $\hat{a}^\dagger$, $\hat{a}$ are bosonic creation and destruction operators, respectively. According to a simple recipe [29], this master equation can be converted to the corresponding Schwinger-Keldysh action

$$S[\boldsymbol{\phi}] = S[\phi_\pm^*, \phi_\pm] = \int_{t_0}^t \mathrm{d}\bar{t}\left[\phi_+^* (\mathrm{i}\partial_{\bar{t}} - \omega_0 + \mathrm{i}\lambda/2)\phi_+ - \phi_-^*(\mathrm{i}\partial_{\bar{t}} - \omega_0 - \mathrm{i}\lambda/2)\phi_- - \mathrm{i}\lambda\phi_+\phi_-^*\right], \tag{18}$$

where $\phi_\pm^*$ denote the complex conjugate fields. In contrast to closed systems, this action couples the forward and backward propagating branches of the Schwinger-Keldysh contour already on the level of non-interacting particles, for which reason it is convenient to formulate the problem on the level of fields as described by the action (18) and originally envisaged by Schwinger [26]. After performing the Keldysh rotation [28]

$$\phi_\pm = \frac{1}{\sqrt{2}}(\phi \pm \hat{\phi}), \tag{19}$$

this action reads

$$S[\phi^*, \phi, \hat{\phi}^*, \hat{\phi}] = \int_{t_0}^t \mathrm{d}\bar{t}\left[\hat{\phi}^*(\mathrm{i}\partial_{\bar{t}} - \omega_0 + \mathrm{i}\lambda/2)\phi + \phi^*(\mathrm{i}\partial_{\bar{t}} - \omega_0 + \mathrm{i}\lambda/2)\hat{\phi} + \mathrm{i}\lambda\hat{\phi}^*\hat{\phi}\right], \tag{20}$$

where $\phi$ is the so-called classical field and $\hat{\phi}$ the so-called response or quantum field, sometimes denoted by $\phi_q$.

To build the bridge to classical non-equilibrium systems, it is now instructive to realise that up to an integration by parts, this is identical to the action functional of the so-called Martin-Siggia-Rose (MSR) [36] path integral (introduced by Janssen [37] and De Dominicis [38]) belonging to the stochastic differential equations derivable from Eq. (17) via phase-space methods [39]. If we introduce quadrature variables $(x, p)$ by a change of variables, and set $\omega_0 = 0$ and $\lambda = 2\theta$, we effectively recover a "double copy" of a special case of the classical Ornstein-Uhlenbeck stochastic process through the action

$$S[p, x, \hat{p}, \hat{x}] = \int_{t_0}^t \mathrm{d}\bar{t}\left[\hat{p}(\partial_{\bar{t}}p + \theta p) + \hat{x}(\partial_{\bar{t}}x + \theta x) + \mathrm{i}\theta(\hat{p}^2 + \hat{x}^2)/4\right]. \tag{21}$$

The general Ornstein-Uhlenbeck process is described by the MSR action

$$S[x, \hat{x}] = \int_{t_0}^{t} d\bar{t} \left[ \hat{x} \left( \partial_{\bar{t}} x + \theta x \right) + iD\hat{x}^2/2 \right]. \tag{22}$$

Hence the Schwinger-Keldysh formalism in its most general form also comprises classical stochastic processes, thus considerably widening the scope of applicability of our algorithm, as will be illustrated in Section 4.2.

Note, however, that the analogy between Eq. (20) and a classical action breaks down for interacting quantum systems. For instance, an interaction term such as $\hat{H}_{\text{int}} \sim \hat{a}^\dagger \hat{a}^\dagger \hat{a} \hat{a}$ results in an equation for the Wigner phase-space distribution with derivatives beyond second order, which is thus not of Fokker-Planck type [39]. In Schwinger-Keldysh field theory, $\hat{H}_{\text{int}}$ in turn leads to non-classical vertices, e.g. $\phi^* \hat{\phi}^* \hat{\phi} \hat{\phi}$, which are no longer Gaussian in the response fields. Detailed discussions of the differences between a classical approximation neglecting these vertices, which on the level of the self-energy essentially amounts to dropping contributions from the spectral function, and the full quantum case can be found in Refs. [2,35] for closed systems, i.e. for $\lambda = 0$ in Eq. (20).

# 3 Numerical Solution of Kadanoff-Baym Equations

The computation of solutions to the KB equations consists formally in finding numerical solutions to an integro-differential equation of the form

$$i\partial_t g(t, t') = h_0(t) g(t, t') + \int_{\mathcal{D}} d\bar{t} \, K(t, \bar{t}) g(\bar{t}, t'), \tag{23}$$

which, together with its adjoint, spans the entire $(t, t')$ plane. Note that $g(t, t')$ and the kernel $K(t, t')$ are assumed to be either skew-Hermitian or symmetric with respect to their arguments. While at first glance Eq. (23) may look like a Fredholm integral equation [58], in physical systems the integrals $\int_{\mathcal{D}} d\bar{t}$ are always reduced to Volterra form, i.e. $\mathcal{D} = [t_0, t]$ or $\mathcal{D} = [t_0, t']$ (cf. Eqs. (16)), the deeper reason for this being causality. Since $K$ is usually a functional of $g$, Eq. (23) belongs to the class of generic non-linear Volterra integro-differential equations (VIDE). For the rest of the analysis, we assume that the integral kernel is smooth and non-singular, as the converse is rarely encountered in the class of physical problems considered here and would require problem-dependent modifications of the quadrature rules to be properly accounted for [59].

The fact that the VIDE (23) is defined on a two-dimensional domain has not only obfuscated its analysis, it has also impeded a direct application of most existing numerical algorithms, which have largely been focused on univariate VIDEs. In the following sections, we present an appropriate discretisation scheme, allowing us to apply general linear methods in solving the KB equations, as well as an exposition on the variable Adams method, our preferred multi-step method for solving these equations.

## 3.1 Stepping Scheme for Kadanoff-Baym Equations

Due to the causal structure of the Volterra initial-value problem, the KB equation at the point $(t, t')$ is only dependent on time arguments smaller or equal to $(t, t')$. By taking the Cartesian product of a (non-equidistant) one-dimensional grid

$$\mathcal{T} := \{t_0 < t_1 < \cdots < t_i < \cdots < t_N\}, \tag{24}$$

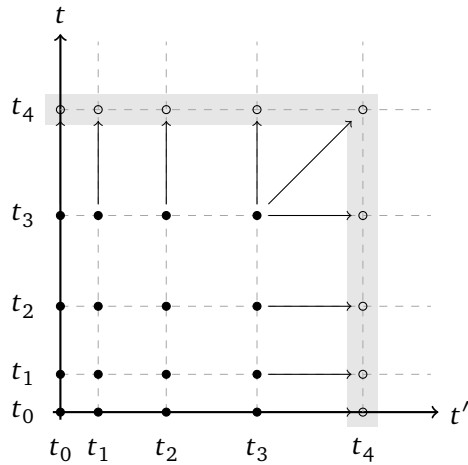

Figure 3: Time-stepping procedure for Eq. (10).

with itself, a symmetric mesh $\mathcal{T} \times \mathcal{T} = \left\{ (t, t') \,|\, t \in \mathcal{T}, t' \in \mathcal{T} \right\}$ for the two-time domain is obtained. Within such a discretisation, the time-stepping procedure can be regarded as a "fan-like" stepping in the symmetric two-time mesh, as depicted in Fig. 3. Accordingly, this can be understood as a system of *univariate*, vector-valued differential equations

$$
\begin{aligned}
\mathrm{i}\partial_{t_i}\mathbf{g}^v(t_i) &= h(t_i)\mathbf{g}^v(t_i) + \big(\mathbf{K} \circ \mathbf{g}\big)^v(t_i) && \text{(vertical step)}, \\
-\mathrm{i}\partial_{t_i}\mathbf{g}^h(t_i) &= \mathbf{g}^h(t_i)h(t_i)^\dagger + \big(\mathbf{g} \circ \mathbf{K}\big)^h(t_i) && \text{(horizontal step)}, \\
\mathrm{i}\partial_{t_i}\mathbf{g}^d(t_i) &= h(t_i)\mathbf{g}^d(t_i) - \mathbf{g}^d(t_i)h(t_i)^\dagger + \big(\mathbf{K} \circ \mathbf{g} - \mathbf{g} \circ \mathbf{K}\big)^d(t_i) && \text{(diagonal step)},
\end{aligned} \tag{25}
$$

where

$$
\begin{aligned}
\mathbf{g}^v(t_i) &= \left[ g(t_i, t_0), g(t_i, t_1), \ldots, g(t_i, t_i) \right], \\
\mathbf{g}^h(t_i) &= \left[ g(t_0, t_i), g(t_1, t_i), \ldots, g(t_i, t_i) \right], \\
\mathbf{g}^d(t_i) &= \left[ g(t_i, t_i) \right],
\end{aligned} \tag{26}
$$

and $\circ$ denotes the element-wise Volterra integration

$$
(\mathbf{A} \circ \mathbf{B})^v(t_i) = \left[ \int_{\mathcal{D}} \mathrm{d}\bar{t}\, A(t_i, \bar{t}) B(\bar{t}, t_0), \int_{\mathcal{D}} \mathrm{d}\bar{t}\, A(t_i, \bar{t}) B(\bar{t}, t_1), \ldots, \int_{\mathcal{D}} \mathrm{d}\bar{t}\, A(t_i, \bar{t}) B(\bar{t}, t_i) \right], \tag{27}
$$

with analogous definitions for the $h$ and $d$ components.

KB equations are set apart from univariate ordinary differential equations (ODEs) or VIDEs by the fact that their dimension grows with each time-step — the size of $\mathbf{g}^v(t)$ and $\mathbf{g}^d(t)$ grows by one when stepping from $t_i$ to $t_{i+1}$. This requires a continued resizing of the equations and is one reason why such equations are not straightforwardly compatible with the extensive amount of available ODE solvers. Moreover, unlike population-growth problems, for example, where the size of the equations may also grow with time, the new equations that are added when solving KB equations have a "past". This can be visualised via Fig. 3 by noting that, for example, when stepping vertically or horizontally from $g(t_4, t_4)$, the right-hand side of the differential equations for the new elements in $\mathbf{g}^v(t)|_{t=t_4}$ and $\mathbf{g}^h(t)|_{t=t_4}$ involve in general non-zero terms at times $t < t_4$. For multi-step methods, in particular, this may necessitate additional care (cf. Section 3.2.1).

Viewing the KB integration procedure effectively as a one-time ODE problem has two main benefits: First, it opens up the possibility of applying virtually any general linear method

to solve KB equations. And second, additional one-time functions such as mean-fields (first cumulants) can be solved simultaneously and in a unified manner, which allows for direct method implementations with well-defined local error estimations.

## 3.2  Univariate Volterra Integro-Differential Equations

Following the structure presented in Eqs. (25), we now focus on a univariate non-linear VIDE in standard form, i.e.

$$y'(t) = F[t, y(t)] + \int_{t_0}^{t} \mathrm{d}s\, K[t, s, y(s)], \tag{28}$$

which can also be seen as a system of two equations, of which one is an ordinary differential equation and the other a Volterra integral equation,

$$\begin{aligned} y'(t) &= F[t, y(t)] + z(t), \\ z(t) &= \int_{t_0}^{t} \mathrm{d}s\, K[t, s, y(s)], \end{aligned} \tag{29}$$

subject to the initial condition

$$y(t_0) = y_0. \tag{30}$$

In some cases, it is possible to solve such equations with analytic methods [58], yet this usually requires the integral kernel to have properties such as linearity, i.e. $K[t, s, y(s)] = K(t, s)\, y(s)$, which is not the case for most physical systems of interest. Hence, we must resort to discrete methods.

While there are many methods one can employ to solve ODEs, *a priori* there is no best method. Its choice strongly depends on factors such as stiffness, desired accuracy and function evaluation cost. We chose to employ a variable order, variable step size Adams (predictor-corrector) method which provides a good trade-off between cost (two function evaluations per step) and overall accuracy, even when the number of equations is very large, as is indeed the case with KB equations, where the number of equations roughly equals the dimension of $G(t_0, t_0)$ times the number of time-steps.

In methods based on integration, Eq. (29) is integrated from $t_n$ to $t_{n+1}$

$$y(t_{n+1}) = y(t_n) + \int_{t_n}^{t_{n+1}} \mathrm{d}s\, \{F[s, y(s)] + z(s)\}, \tag{31}$$

and the integrals are then evaluated with interpolating quadrature formulas. Here it becomes clear that the main computational bottleneck in solving these equations is in the computation of $z(t)$, which can be evaluated with a so-called direct quadrature method

$$z(t_n) = \int_{t_0}^{t} \mathrm{d}s\, K[t_n, s, y(s)] = \sum_{\ell=0}^{n-1} \int_{t_\ell}^{t_{\ell+1}} \mathrm{d}s\, K[t_n, s, y(s)]. \tag{32}$$

Nonetheless, it is possible to further differentiate $z(t)$ and treat $\{y'(t), z'(t)\}$ as a system of coupled differential equations [60], which would be more suitable in cases where the integral equation is stiff $(-\partial K/\partial y \gg 1)$ [61]. We opted for the former due to its simpler implementation and the fact that most physical systems of interest do not satisfy such stiffness criterion.

### 3.2.1 Variable Adams method

The variable Adams method [62] is a predictor-corrector scheme where the integrand of Eq. (31) is approximated by a Newton polynomial, that is, an interpolation polynomial with previously computed points. A prediction $y_{n+1}^*$ for the solution of $y(t_{n+1})$ – note that here $^*$ denotes the prediction, not complex conjugation – is obtained via an explicit method with a $(k-1)$-th order polynomial

$$y_{n+1}^* = y_n + \int_{t_n}^{t_{n+1}} ds \sum_{j=0}^{k-1} \left[ \prod_{i=0}^{j-1} (s - t_{n-i}) \right] \delta^j \{ F[t_n, y(t_n)] + z(t_n) \}, \qquad (33)$$

and the divided differences are defined recursively as

$$\delta^0 F[t_\ell, y(t_\ell)] = F[t_\ell, y(t_\ell)],$$
$$\delta^j F[t_\ell, y(t_\ell)] = \frac{\delta^{j-1} F[t_\ell, y(t_\ell)] - \delta^{j-1} F[t_{\ell-1}, y(t_{\ell-1})]}{t_\ell - t_{\ell-j}}. \qquad (34)$$

The prediction for $y(t_{n+1})$ is now corrected via an implicit method, where the $k$-th order interpolation polynomial of the integrand makes use of the predicted value $y_{n+1}^*$:

$$y_{n+1} = y_{n+1}^* + \int_{t_n}^{t_{n+1}} ds \left[ \prod_{i=0}^{k-1} (s - t_{n-i}) \right] \delta^k \{ F[t_{n+1}, y(t_{n+1})] + z(t_{n+1}) \}. \qquad (35)$$

The integrals in Eq. (32) can be evaluated in the same predictor-corrector manner:

$$z_n^* = \sum_{\ell=0}^{n-1} \int_{t_\ell}^{t_{\ell+1}} ds \sum_{j=0}^{k-1} \left[ \prod_{i=0}^{j-1} (s - t_{\ell-i}) \right] \delta^j K_n[t_\ell, y(t_\ell)],$$
$$z_n = z_n^* + \sum_{\ell=0}^{n-1} \int_{t_\ell}^{t_{\ell+1}} ds \left[ \prod_{i=0}^{k-1} (s - t_{\ell-i}) \right] \delta^k K_n[t_{\ell+1}, y(t_{\ell+1})], \qquad (36)$$

with divided differences defined as

$$\delta^0 K_n[t_\ell, y(t_\ell)] = K[t_n, t_\ell, y(t_\ell)],$$
$$\delta^j K_n[t_\ell, y(t_\ell)] = \frac{\delta^{j-1} K_n[t_\ell, y(t_\ell)] - \delta^{j-1} K_n[t_{\ell-1}, y(t_{\ell-1})]}{t_\ell - t_{\ell-j}}. \qquad (37)$$

The main difficulties when evaluating the predictor-corrector Eqs. (33) and (35) are that it is challenging to obtain a closed formula for the integrals, and that it is algorithmically expensive to calculate the divided differences via recursive formulas (Eq. (34)). While for equidistant time grids the equations find a simple and compact form [62], in the non-equidistant case the expressions rapidly become convoluted and complicated to implement. These problems can be circumvented by recurrence formulas [62], which make the evaluation of the integrals and $j$-th derivatives more efficient.

In between time steps, an estimate of the local truncation error can be obtained by computing $\tilde{y}_{n+1} - y_{n+1}$, where $\tilde{y}_{n+1}$ is the result of the implicit step using a $(k+1)$-th order formula. It is assumed that as $k \to \infty$, the error approaches zero (in which case the integral quadrature formula is said to be *convergent*). A measure of this error satisfying specific tolerances is obtained via

$$le_k(n+1) := \frac{\tilde{y}_{n+1} - y_{n+1}}{\texttt{atol} + \texttt{rtol} \cdot \max(|y_n|, |y_{n+1}|)}, \qquad (38)$$

for which the integration step is accepted if

$$\|le_k(n+1)\|_p \leq 1, \tag{39}$$

and the norm is defined as

$$\|x\|_p = \left(\frac{1}{n}\sum_i^n |x^i|^p\right)^{\frac{1}{p}}, \tag{40}$$

where typically $p = 2$. Given this acceptance criterion, the roles of the tolerances `rtol` and `atol` in Eq. (38) can be better understood considering them separately under the infinity-norm. In this scenario, $-\log_{10}\texttt{rtol}$ controls the minimum number of correct digits between time steps, while `atol` is a threshold for the magnitude of the elements of $y$ for which the minimum number of correct digits is guaranteed. This local error is then used to adjust both the step size $h_n := (t_{n+1} - t_n)$ and the order $k$. The next time step is chosen as the largest possible step that still satisfies the local error being $\lesssim 1$. Given the current local error $\|le_k(n+1)\| \simeq Ch_n^{k+1}$ for some constant $C$, and assuming that the subsequent error is maximal, i.e. $\|le_k(n+2)\| \simeq Ch_{n+1}^{k+1} \approx 1$, the next time step can be chosen optimally as [62]

$$h_{n+1} = h_n \|le_k(n+1)\|_p^{-\frac{1}{k+1}}. \tag{41}$$

Obtaining the optimal order $k$ is slightly more involved and we refer the reader to Ref. [62] for an excellent and self-contained explanation of heuristic mechanisms for order selection. Regardless of the order $k$, the number of required function evaluations per time step is constant, hence $k$ is ideally set to a large value ($\gtrsim 5$) such that the integrator can take larger steps and the overall computational cost is reduced.

## 3.3 Volterra Integral Equations of the Second Kind

More elaborate self-energy approximations ($GW$, $T$-matrix [16, 19, 63], $1/\mathcal{N}$ [2, 3]), which comprise of resummations of particular classes of diagrams, require the solution of Volterra integral equations of the second kind [64]:

$$I(t, t') = \Phi(t, t') - \int_{\mathcal{C}} d\bar{t}\ \Phi(t, \bar{t})I(\bar{t}, t'). \tag{42}$$

In the mentioned self-energy approximations, the kernel $K(t, t')$ of Eq. (23) then typically depends linearly on $I(t, t')$ as shown in Eq. (62), and $\Phi(t, t')$ is a function of $g(t, t')$.

There are several ways of solving Eq. (42): by inversion of the triangular system of equations obtained when discretizing in the same manner as in Eq. (25), by reduction to a VIDE through differentiation, or by iteration of the equation [53]. Since Eq. (42) has to be solved simultaneously with Eq. (23), reducing it to a VIDE would be ideal, yet this generally results in stiff equations [60] for which the variable Adams method (Section 3.2.1) is not appropriate. To achieve congruity with the method previously presented, we solve Eq. (42) iteratively at every predictor and corrector step, i.e. following the same evolution procedure as depicted in Fig. 3.

## 3.4 Complexity Reduction from Symmetries and Physical Properties

Leveraging symmetries and other physical properties of a system can significantly reduce the computational effort on top of what can be achieved by adaptive time-stepping. Here we briefly discuss two aspects which are relevant in this respect.

### 3.4.1 Symmetries in the Two-Time Domain

Apart from the symmetries of the Hamiltonian, the two-time Green functions encountered in quantum and classical systems possess symmetries in the two-time domain $(t, t')$. For example, in the quantum case our numerical implementations of are based on the greater and lesser Green functions that are skew-Hermitian in time,

$$\left[ G^{\lessgtr}(t, t') \right]^{\dagger} = -G^{\lessgtr}(t', t). \tag{43}$$

Hence, the solutions are fully determined by either the upper- or lower-triangular elements, which essentially cuts in half the number of equations by requiring only the integration of $\mathbf{G}^d$ and either $\mathbf{G}^v$ or $\mathbf{G}^h$. Similar relations hold true for classical stochastic processes, however with different symmetry relations, which we discuss in Eq. (82) of Section 4.2.1.

### 3.4.2 Memory truncation

The clustering decomposition principle [65] ensures that at a large-enough time separation of the physical operators, any $n$-point function factorizes. In terms of *connected* 2-point functions this has the signature of an exponential or power-law *decay* in the relative-time direction (s. Appendix A), for massive and massless fields, respectively [66]. This principle should hold for any stable, long-lived state, an example being thermalised systems [67] described by a Gibbs ensemble. This effect is similarly present in physical systems connected to some kind of reservoir (e.g. as in open quantum systems [34] or quantum impurity systems described by dynamical mean-field theory [48]). The VIDE can hence often be approximated by a Volterra *delay*-integro-differential equation with

$$z(t) = \int_{t-\tau}^{t} \mathrm{d}s \, K\left[t, s, y(s)\right], \tag{44}$$

where $\tau$ is some cut-off time. This is rooted in the fact that the physical Green functions in such systems display long-time decay and thus

$$\left\| \int_{t-\tau}^{t} \mathrm{d}s \, K[t, s, y(s)] \right\| \gg \left\| \int_{t_0}^{t-\tau} \mathrm{d}s \, K[t, s, y(s)] \right\|. \tag{45}$$

Since one bottleneck when solving KB equations is in the evaluation of the integrals, introducing a cut-off time can dramatically reduce the computational complexity from $\mathcal{O}\left(n^2 k(n)\right)$ to $\mathcal{O}\left(n^2 k(N_\tau)\right)$ where $n$ denotes the number of time-steps, $N_\tau$ the number of time points in the interval $[t-\tau, t]$ and $k$ is the complexity of integrating the kernel as a function of the number of required time points. Moreover, these grid points can then also be excluded from future time evolution, which further reduces the overall complexity to $\mathcal{O}\left(n N_\tau k(N_\tau)\right)$. This point and its relation to the generalised Kadanoff-Baym ansatz [49] are taken up again in our discussion of the Fermi-Hubbard model (cf. Section 4.1.3 and Fig. 10).

The sensitivity of the algorithm to values off the two-time diagonal can be explicitly adjusted via the parameter `atol`, irrespective of the nature of the decay of the Green functions away from the diagonal. For rapid (exponential) decay, e.g. in a driven system, a given value of this parameter will lead to a small number of grid points. For slow (algebraic) decay, the same tolerances will result in a larger number of grid points.

## 4 Numerical Examples

After having discussed the theoretical framework of field theory and the adaptive algorithm in the previous sections, we now turn to the numerical solution of a number of specific benchmark

problems. The present section is organised into two parts, the first of which covers quantum systems in Section 4.1, while the second part Section 4.2 deals with classical stochastic processes.

## 4.1 Quantum Systems

We begin with an error analysis and a comparison between fixed and adaptive methods for the tight-binding Hamiltonian in Section 4.1.1. Subsequently, we dive into interacting quantum systems, considering a bosonic mixture in Section 4.1.2, and the Fermi-Hubbard model in Section 4.1.3. The section on quantum dynamics is rounded off with a detailed exposition of how to apply the Schwinger-Keldysh formalism to open quantum systems in Section 4.1.4. We note that for the examples presented without analytical solution, the numerical solutions were computed such that no appreciable difference was observed when selecting stricter tolerances.

### 4.1.1 One-Dimensional Tight-Binding Model

As the tight-binding model allows for a straightforward analytical solution, while at the same time being the non-interacting limit of many non-trivial strongly-correlated matter models, we take it as the basis for an investigation of the error scaling of the adaptive algorithm. For the model Hamiltonian, we set

$$\hat{H} = \sum_{i=1}^{L} \varepsilon_i \hat{c}_i^\dagger \hat{c}_i + J \sum_{\langle i,j \rangle} \left( \hat{c}_i^\dagger \hat{c}_j + \hat{c}_j^\dagger \hat{c}_i \right), \tag{46}$$

where the energy on site $i$ and the kinetic energy are parameterised by $\varepsilon_i$ and $J$, respectively. For simplicity we ignore spin degrees of freedom. In terms of fermionic creation and annihilation operators $\{\hat{c}_i, \hat{c}_j^\dagger\} = 1$, the greater and lesser Green functions are defined as

$$\begin{aligned}
\left[ \mathbf{G}^>(t,t') \right]_{ij} = G_{ij}^>(t,t') &= -i \langle \hat{c}_i(t) \hat{c}_j^\dagger(t') \rangle, \\
\left[ \mathbf{G}^<(t,t') \right]_{ij} = G_{ij}^<(t,t') &= \phantom{-}i \langle \hat{c}_j^\dagger(t') \hat{c}_i(t) \rangle.
\end{aligned} \tag{47}$$

Their equations of motion are given by

$$\begin{aligned}
i\partial_t \mathbf{G}^{\lessgtr}(t,t') &= \mathbf{H} \mathbf{G}^{\lessgtr}(t,t'), \\
i\partial_T \mathbf{G}^{\lessgtr}(T,0)_W &= [\mathbf{H}, \mathbf{G}^{\lessgtr}(T,0)_W],
\end{aligned} \tag{48}$$

for the vertical time directions $t$ and for the centre-of-mass, i.e., diagonal time direction $T = (t + t')/2$, respectively. The Wigner-transformed Green functions $\mathbf{G}^{\lessgtr}(T,0)_W$ are defined in Appendix A. For a one-dimensional tight-binding chain of length $L$ in the position basis, the Hermitian matrix $\mathbf{H}$ is tridiagonal and reads

$$\mathbf{H} = \begin{pmatrix} \varepsilon_1 & J & & \\ J & \ddots & \ddots & \\ & \ddots & \ddots & J \\ & & J & \varepsilon_L \end{pmatrix}. \tag{49}$$

The analytical solution is found via matrix exponentials from

$$\mathcal{G}^{\lessgtr}(t,t') = e^{-iHt} \mathcal{G}^{\lessgtr}(0,0) e^{iHt'}, \tag{50}$$

where $\mathcal{G}^{\lessgtr}(0,0)$ are the initial conditions. We now set $L = 2$ for simplicity and compare our numerical results with the analytical ones following from Eq. (50). A key result of this section

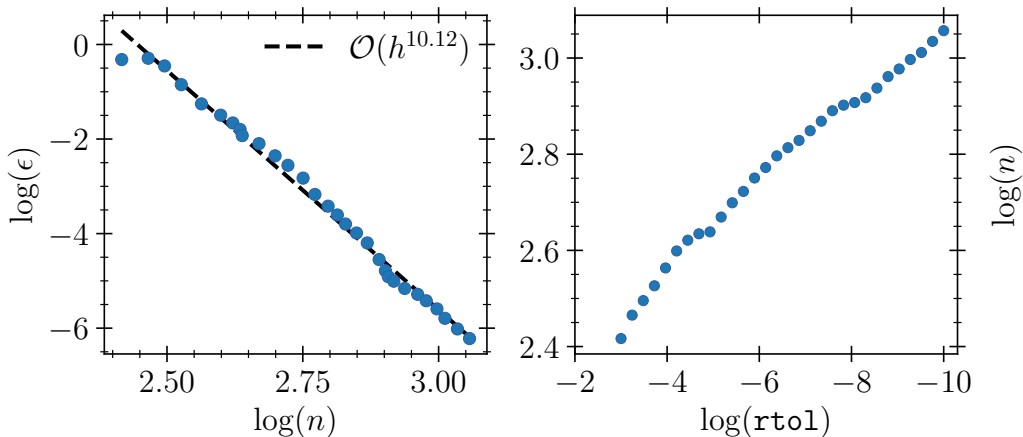

Figure 4: Absolute error $\epsilon$ for the two-site tight-binding model Eq. (46) as a function of the number of time-steps $n$ and the number of time-steps as function of the tolerances, with $\mathtt{atol} = 10^{-2}\,\mathtt{rtol}$. The time boundaries were kept fixed (with $Jt_{\mathrm{final}} = 5$) and the maximum algorithm order chosen was $k_{\mathrm{max}} = 9$. The black dashed line is a linear fit corroborating the error scaling as $\mathcal{O}(h^{k_{\mathrm{max}}+1})$.

is shown in the left panel of Fig. 4, where the total evolution time interval, $t, t' \in [0, t_{\mathrm{final}}]$, is kept fixed and it is studied how the overall error $\epsilon = \|G^< - \mathcal{G}^<\|_2$ — using the norm of Eq. (40) — scales with the number of time-steps $n$. The latter is controlled via the tolerances $\mathtt{rtol}$ and $\mathtt{atol}$ (s. Fig. 4). Assuming the (adaptive) time-steps to be approximately equal, $h \approx t_{\mathrm{final}}/(n-1)$, we find a scaling exponent in agreement with theory [62]: $\mathcal{O}(h^{k_{\mathrm{max}}+1})$ where $k_{\mathrm{max}}$ is the maximum order of the integrator. The other key result is shown in Fig. 5, where the adaptive stepping is compared with a semi-fixed and a fixed stepping scheme. The difference between adaptive and semi-fixed is that the latter has a deliberate maximum step size $h_{\mathrm{max}}$. This is because for a fixed stepping, with constant $h$, too much error is accumulated at the beginning of the time integration. Comparing the adaptive and semi-fixed schemes, the algorithm achieves a smaller error with a fraction of the time-steps of a fixed stepping

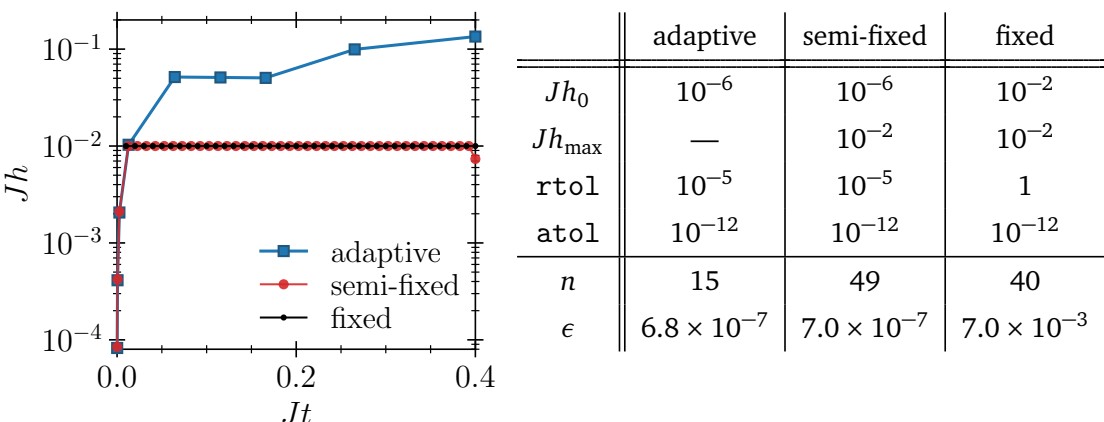

|  | adaptive | semi-fixed | fixed |
|---|---|---|---|
| $Jh_0$ | $10^{-6}$ | $10^{-6}$ | $10^{-2}$ |
| $Jh_{\mathrm{max}}$ | — | $10^{-2}$ | $10^{-2}$ |
| $\mathtt{rtol}$ | $10^{-5}$ | $10^{-5}$ | $1$ |
| $\mathtt{atol}$ | $10^{-12}$ | $10^{-12}$ | $10^{-12}$ |
| $n$ | 15 | 49 | 40 |
| $\epsilon$ | $6.8 \times 10^{-7}$ | $7.0 \times 10^{-7}$ | $7.0 \times 10^{-3}$ |

Figure 5 & Table 1: A comparison between adaptive and fixed stepping schemes for the two-site tight-binding model discussed in Section 4.1.1. The effective step size is denoted by $h$, the maximal allowed step size by $h_{\mathrm{max}}$, and the tolerances $\mathtt{rtol}$ and $\mathtt{atol}$ are discussed in Section 3.2.1. The adaptive scheme reduces the number of points $n$ considerably, while also providing the smallest error.

scheme. This efficiency leads to a considerable advantage when tackling numerically expensive interacting problems with long evolution times.

### 4.1.2 Bosonic Mixture with Excitation Transfer

Ultracold bosonic atoms exhibiting Bose-Einstein condensation have long been an important experimental platform to investigate fundamental physics such as matter-wave interference [70] and quantum thermalisation [71, 72]. Binary mixtures of two condensate types [73], in turn, give rise to a number of interesting collective effects like demixing [74] or breathing modes [75]. Here, we apply the adaptive algorithm to a binary bosonic mixture on a lattice of length $L$, modelled by the Hamiltonian

$$\hat{H} = \omega_0 \sum_{i=1}^{L} \left( \hat{b}_i^\dagger \hat{b}_i - \hat{a}_i^\dagger \hat{a}_i \right) + J \sum_{\langle i,j \rangle} \hat{a}_i^\dagger \hat{b}_i \hat{b}_j^\dagger \hat{a}_j \,, \tag{51}$$

where the bosonic operators $\hat{a}_i$, $\hat{a}_i^\dagger$ ($\hat{b}_i$, $\hat{b}_i^\dagger$) describe first (second) species of atoms on lattice site $i$. The excitation can be transferred between neighbouring sites via the interaction $J$, which induces intra-species particle transport even in the absence of a direct hopping term between the sites. For the purpose of benchmarking the numerics, we do not consider spontaneous symmetry breaking (Bose-Einstein condensation), but focus on a regime with negligible order parameter. Then, if the correlations are prepared to be initially local, the single-particle Green function will remain local for all times, i.e. diagonal in the site index (no particle hopping), since the interaction $J$ does not induce non-local terms in any order of perturbation theory [4]. Nevertheless, non-local correlations of the excitation amplitude will be induced, as seen below. Hence, we define two types of contour-ordered, diagonal Green functions

$$\begin{aligned} \mathcal{A}_{i,j}(t,t') &= -\mathrm{i}\delta_{ij}\langle \mathcal{T}_\mathcal{C} \hat{a}_i(t)\hat{a}_i^\dagger(t') \rangle \,, \\ \mathcal{B}_{i,j}(t,t') &= -\mathrm{i}\delta_{ij}\langle \mathcal{T}_\mathcal{C} \hat{b}_i(t)\hat{b}_i^\dagger(t') \rangle \,. \end{aligned} \tag{52}$$

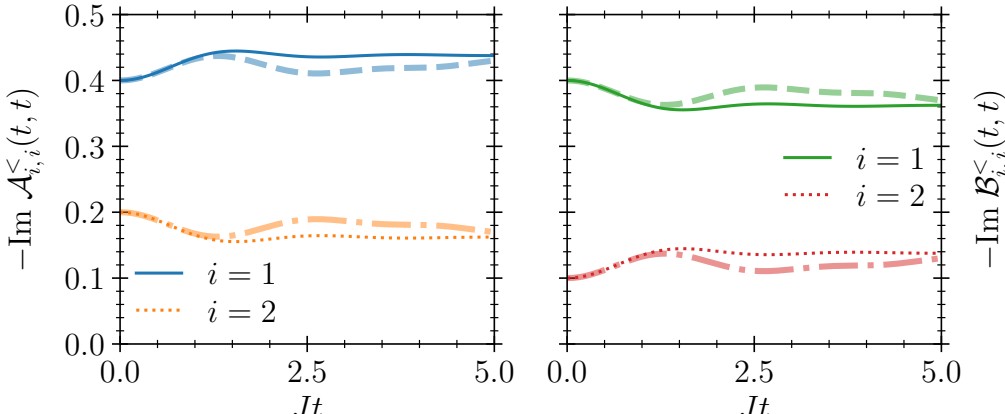

Figure 6: Evolution of the site occupations $-\mathrm{Im}\,\mathcal{A}_{i,i}^<(t,t)$ and $-\mathrm{Im}\,\mathcal{B}_{i,i}^<(t,t)$ for the bosonic mixture Eq. (51), with $\omega_0 = 5J$ and initial conditions $\mathcal{A}^<(0,0) = -\mathrm{i}\,\mathrm{diag}(0.4,0.2)$, $\mathcal{B}^<(0,0) = -\mathrm{i}\,\mathrm{diag}(0.4,0.1)$. Thick lines indicate numerically exact benchmark results obtained from exact diagonalisation with QuTiP [68]. The damping visible in our results is known to arise from the employed second-order approximation and can be improved upon via more advanced diagrammatic expansions [69]. The numerical tolerances are `atol=1e-10` and `rtol=1e-8`.

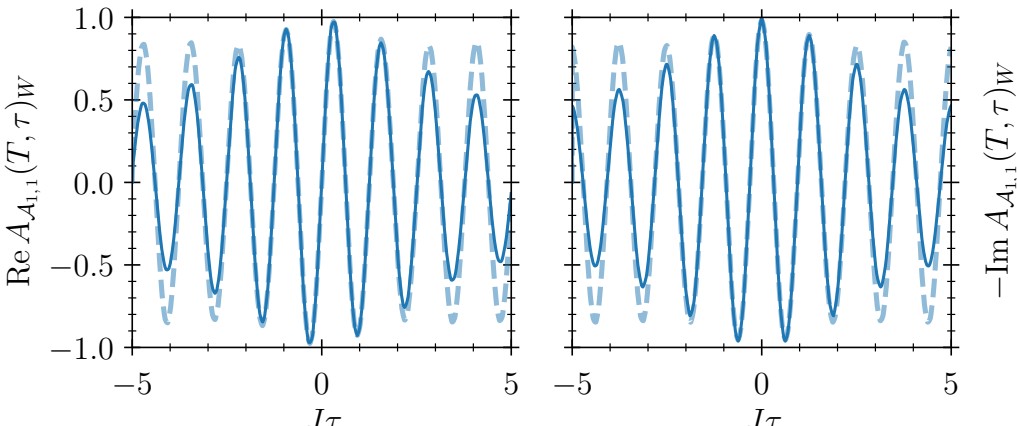

Figure 7: Spectral function $A_{\mathcal{A}_{1,1}}(T,\tau)_W = \left[\mathcal{A}_{1,1}^>(T,\tau)_W - \mathcal{A}_{1,1}^<(T,\tau)_W\right]$ for the bosonic mixture Eq. (51) at centre-of-mass time $JT = 2.5$ (s. Appendix A). Dashed lines again show numerically exact results (cf. Fig. 6).

Since no off-diagonal Green functions appear if they are zero initially, there is no Hartree-Fock contribution to $\Gamma_2$. The lowest-order diagrams are hence of second order in $J$, and the corresponding terms in the 2PI effective action read

$$\Gamma_2[\mathcal{A},\mathcal{B}] = \frac{iJ^2}{2}\sum_{\langle i,j\rangle}\int dt dt'\, \mathcal{A}_{i,i}(t,t')\mathcal{B}_{i,i}(t',t)\mathcal{A}_{j,j}(t',t)\mathcal{B}_{j,j}(t,t'). \tag{53}$$

Following Eq. (9), the self-energy components then become

$$\begin{aligned}
\left[\boldsymbol{\Sigma}_{\mathcal{A}}^{\lessgtr}(t,t')\right]_{i,i} &= -J^2\sum_{j\in\mathcal{N}_i}\mathcal{B}_{i,i}^{\lessgtr}(t,t')\mathcal{A}_{j,j}^{\gtrless}(t,t')\mathcal{B}_{j,j}^{\gtrless}(t',t),\\
\left[\boldsymbol{\Sigma}_{\mathcal{B}}^{\lessgtr}(t,t')\right]_{i,i} &= -J^2\sum_{j\in\mathcal{N}_i}\mathcal{A}_{i,i}^{\lessgtr}(t,t')\mathcal{A}_{j,j}^{\gtrless}(t',t)\mathcal{B}_{j,j}^{\lessgtr}(t,t'),
\end{aligned} \tag{54}$$

where $\mathcal{N}_i$ is the set of nearest neighbours of site $i$. The Dyson equations of motion for the components of the greater and lesser Green functions are finally

$$\begin{aligned}
(i\partial_t + \omega_0 \mathrm{diag}(1,\dots,1))\mathcal{A}^{\lessgtr}(t,t') &= \int_0^t d\bar{t}\left[\boldsymbol{\Sigma}_{\mathcal{A}}^>(t,\bar{t}) - \boldsymbol{\Sigma}_{\mathcal{A}}^<(t,\bar{t})\right]\mathcal{A}^{\lessgtr}(t,t')\\
&\quad + \int_0^{t'} d\bar{t}\,\boldsymbol{\Sigma}_{\mathcal{A}}^{\lessgtr}(t,\bar{t})\left[\mathcal{A}^<(t,t') - \mathcal{A}^>(t,t')\right],\\
(i\partial_t - \omega_0 \mathrm{diag}(1,\dots,1))\mathcal{B}^{\lessgtr}(t,t') &= \int_0^t d\bar{t}\left[\boldsymbol{\Sigma}_{\mathcal{B}}^>(t,\bar{t}) - \boldsymbol{\Sigma}_{\mathcal{B}}^<(t,\bar{t})\right]\mathcal{B}^{\lessgtr}(t,t')\\
&\quad + \int_0^{t'} d\bar{t}\,\boldsymbol{\Sigma}_{\mathcal{B}}^{\lessgtr}(t,\bar{t})\left[\mathcal{B}^<(t,t') - \mathcal{B}^>(t,t')\right].
\end{aligned} \tag{55}$$

Exemplary results for a simple one-dimensional lattice with $L = 2$ are shown in Fig. 6 along the time diagonal and in Fig. 7 along the relative-time axis (s. Appendix A). Fig. 6 shows there is indeed intra-species particle transport even without direct single-particle hopping. The $\tau$ dynamics shown in Fig. 7 indicate that the approximation gives a good value for the oscillation frequency, while the relaxation rate is over-estimated (we emphasise that this is a result of the diagrammatic approximation [69] rather than of our simulation).

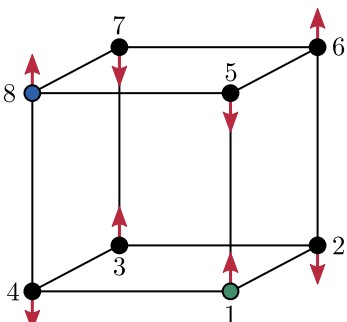

Figure 8: A schematic representation of the three-dimensional Fermi-Hubbard lattice. The (red) arrows give a qualitative indication of the initial spin configuration, which we use to initialise the charge and spin dynamics shown in Fig. 9 for sites 1 and 8.

### 4.1.3 Fermi-Hubbard Model

The Fermi-Hubbard model is one of the central models of quantum many-body physics, able to capture the rich phases of matter displayed in strongly-correlated electronic materials. Albeit seemingly simple in form, its analytical and numerical solutions are difficult to obtain. Let a three-dimensional Fermi-Hubbard lattice of $L = 8$ sites as depicted in Fig. 8, with nearest-neighbour hopping $J$ and on-site repulsive interaction $U$, be described by a Hamiltonian

$$\hat{H} = -J \sum_{\langle i,j \rangle} \sum_{\sigma} \hat{c}_{i,\sigma}^{\dagger} \hat{c}_{i+1,\sigma} + U \sum_{i=1}^{L} \hat{c}_{i,\uparrow}^{\dagger} \hat{c}_{i,\uparrow} \hat{c}_{i,\downarrow}^{\dagger} \hat{c}_{i,\downarrow}, \tag{56}$$

where the fermionic creation and annihilation operators obey $\{\hat{c}_{i,\sigma}, \hat{c}_{j,\sigma'}^{\dagger}\} = \delta_{ij}\delta_{\sigma,\sigma'}$, with $i = 1, \ldots, L$ and $\sigma = \uparrow, \downarrow$. Since numerically exact benchmark results are already difficult to obtain for this model, no such comparison is undertaken here. We introduce the spin-diagonal,

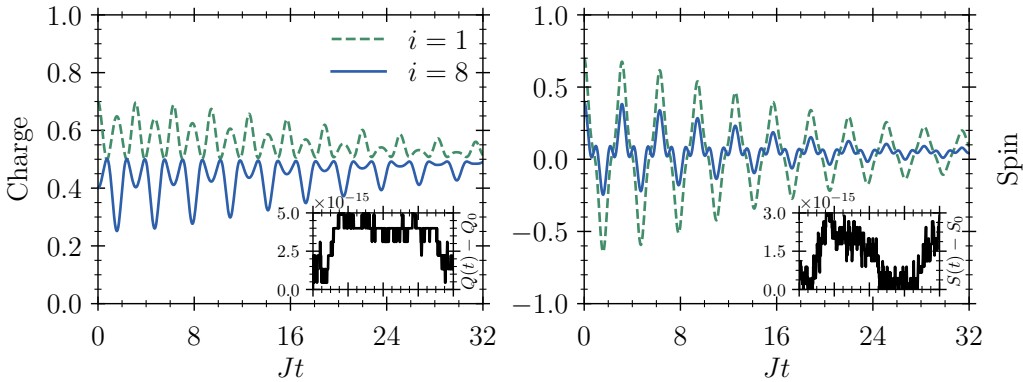

Figure 9: Long-time evolution of the eight-site Fermi-Hubbard lattice (Fig. 8) in second Born approximation, with interaction $U/J = 1/4$ and density-modulated initial conditions $\boldsymbol{G}_{\uparrow}^{<}(0,0) = \mathrm{i}\,\mathrm{diag}(0.7, 0.0, 0.7, 0.0, 0.0, 0.4, 0.0, 0.4)$, $\boldsymbol{G}_{\downarrow}^{<}(0,0) = \mathrm{i}\,\mathrm{diag}(0.0, 0.25, 0.0, 0.25, 0.65, 0.0, 0.65, 0.0)$. The initial total charge is hence $Q_0 = 4$, while for spin we have $S_0 = 0.4$. The insets show that the conservation of the total charge $Q(t)$ and the total spin $S(t)$ is satisfied at machine precision. The on-site charge dynamics is described by $\sum_{\sigma} \mathrm{Im} G_{ii,\sigma}^{<}(t,t)$, whereas the spin dynamics is defined as $\mathrm{Im} G_{ii,\uparrow}^{<}(t,t) - \mathrm{Im} G_{ii,\downarrow}^{<}(t,t)$. The numerical tolerances are `atol=1e-8` and `rtol=1e-6`.

contour-ordered Green functions

$$G_{ij,\sigma}(t,t') = -i\langle \mathcal{T}_\mathcal{C} \hat{c}_{i,\sigma}(t) \hat{c}^\dagger_{j,\sigma}(t') \rangle, \tag{57}$$

in terms of which the interacting part of the 2PI effective action becomes

$$\begin{aligned}
\Gamma_2[\boldsymbol{G}_{\uparrow,\downarrow}] = &-U \int dt \sum_{i=1}^{L} G_{ii,\uparrow}(t,t) G_{ii,\downarrow}(t,t) \\
&+ \frac{i}{2} U^2 \int dt dt' \sum_{i,j=1}^{L} G_{ij,\uparrow}(t,t') G_{ji,\uparrow}(t',t) G_{ij,\downarrow}(t,t') G_{ji,\downarrow}(t',t).
\end{aligned} \tag{58}$$

This gives rise to two contributions to the self-energy. The local one is of Hartree-Fock form and reads

$$\begin{aligned}
\Sigma^{\mathrm{HF}}_{ij,\uparrow}(t,t') &= i\delta_{ij}\delta(t-t') G^{<}_{ii,\downarrow}(t,t), \\
\Sigma^{\mathrm{HF}}_{ij,\downarrow}(t,t') &= i\delta_{ij}\delta(t-t') G^{<}_{ii,\uparrow}(t,t),
\end{aligned} \tag{59}$$

where the operator ordering in the Hamiltonian determines the local functions $G_{ii,\sigma}$ on the right-hand side to be lesser Green functions. To the next order in the interaction parameter $U$, we obtain the non-local self-energy contributions

$$\begin{aligned}
\Sigma^{\mathrm{2B}}_{ij,\uparrow}(t,t') &= U^2 G_{ij,\uparrow}(t,t') G_{ij,\downarrow}(t,t') G_{ji,\downarrow}(t',t), \\
\Sigma^{\mathrm{2B}}_{ij,\downarrow}(t,t') &= U^2 G_{ij,\downarrow}(t,t') G_{ij,\uparrow}(t,t') G_{ji,\uparrow}(t',t),
\end{aligned} \tag{60}$$

which are also known as the second Born approximation. Following Section 2.2.1, it is then straightforward to obtain the equations of motion in the form of Eqs. (16). Exemplary results of our numerical solution of the resulting equations are shown in Figs. 9 and 10 for inhomogeneous quarter filling, i.e. a total charge of $Q_0 = 4$ initially distributed over the sites. Total charge and spin conservation (s. insets of Fig. 9) are satisfied at machine precision, warranting the fulfilment of the conserving approximation inherent in the construction of the self-energies via $\Gamma_2$ (which acts as the $\Phi$ functional in a "$\Phi$-derivable" scheme). For the investigated final time $Jt = 32$ and at the given tolerances, this simulation runs quickly on a present-day conventional laptop. Fig. 10 shows the spectral function of site 1 on the complete $(t,t')$ mesh,

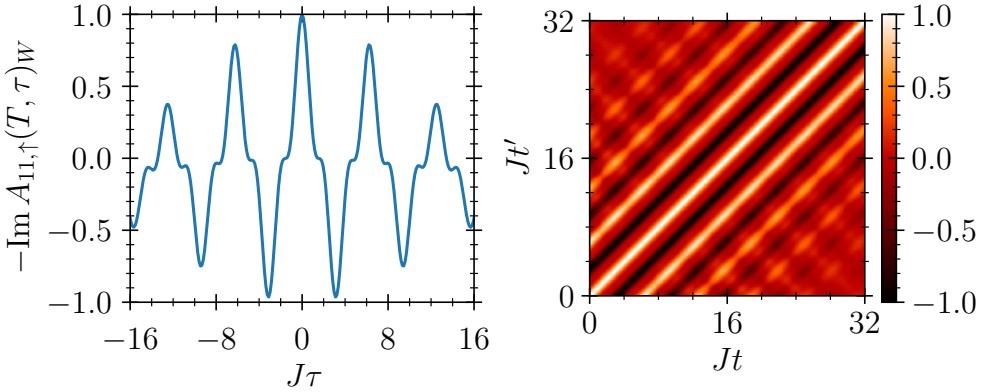

Figure 10: The Fermi-Hubbard spectral function $A_{11,\uparrow}(T,\tau)_W$ at centre-of-mass time $JT = 16$ (left) and $-\operatorname{Im} A_{11,\uparrow}(t,t')$ (right) for settings as in Fig. 9. In the approximation given by Eq. (58), the interaction produces a damping in the $\tau$-direction, orthogonally to the equal-time diagonal.

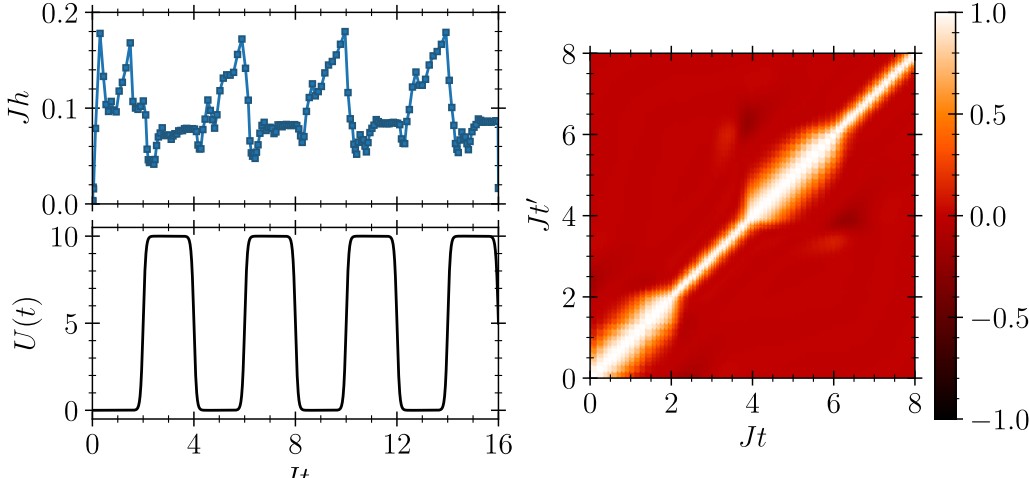

Figure 11: The Fermi-Hubbard spectral function $-\text{Im}\,A_{11,\uparrow}(t,t')$ (right) for the same setting as in Fig. 9, yet with a time-dependent interaction parameter $U(t)$ (bottom left). The (upper left) panel shows the resulting effective step size $h$ for tolerances `atol=1e-5` and `rtol=1e-3`. The right panel does not stretch over the full integration time for better visibility.

as well as its profile in $\tau$-direction. From a numerical perspective, the interesting feature to observe is the interaction-induced damping along the $\tau$ axis, i.e. orthogonally to the equal-time diagonal $t = t'$. As discussed in 3.4.2, this observation holds quite generally for quantum many-body systems [67], and is also related to the generalised Kadanoff-Baym ansatz [49]. To make optimal use of this feature for reducing the memory requirements during the simulation of KB equations, an efficient and straightforward approach is to truncate the computation of the data points at some fixed, problem-dependent distance to the equal-time diagonal [34]. In combination with sparse matrices, this results in a "quasi-linear" scaling in the number of time-steps, thus in principle allowing for long evolution times.

The main strength of our adaptive algorithm can be understood from Fig. 11, which shows a prototypical example of a system with varying time scales. The now time-dependent interaction parameter $U(t)$ is switched on and off periodically, thus inducing, in particular, different decay rates in the relative-time direction. The resulting effective step size $h$ closely follows the development of $U(t)$, relaxing back to larger values when feasible, while quickly contracting again when the interaction is rapidly ramped up. Thus, when dealing with varying time scales, adaptivity appears to be the natural solution. Note that even though the approximation defined by Eq. (58) is not expected to be accurate in the regimes where $U(t)$ is large, this does not affect the validity of our argument regarding adaptivity. Physical examples where adaptivity could be particularly beneficial are systems displaying prethermalisation [76] or the condensation thresholds in photonic condensates out of equilibrium [34, 77, 78], where a short-time evolution with rapid changes is typically followed by a very slow long-time evolution.

For the remainder of this section, we study the system in the $T$-matrix approximation [53, 63], for which the interacting part of the 2PI effective action can be written as

$$\Gamma_2[\boldsymbol{G}_{\uparrow,\downarrow}] = -\text{i}\,\text{Tr}\sum_{n=1}^{\infty}\frac{(-\text{i}U)^n}{n}\int_{\mathcal{C}}\text{d}t_1\cdots\text{d}t_n\left[\prod_{k=1}^{n-1}\boldsymbol{G}_{\uparrow}(t_k,t_{k+1})\circ\boldsymbol{G}_{\downarrow}(t_k,t_{k+1})\right]\boldsymbol{G}_{\uparrow}(t_n,t_1)\circ\boldsymbol{G}_{\downarrow}(t_n,t_1), \quad (61)$$

where truncating the sum at $n = 2$ yields Eq. (58), and $[\boldsymbol{A}\circ\boldsymbol{B}]_{ij} = A_{ij}B_{ij}$. The self-energies,

beyond Hartree-Fock, are then given by

$$
\begin{aligned}
\Sigma_{ij,\uparrow}^{TM}(t,t') &= iU^2 T_{ij}(t,t')G_{ji,\downarrow}(t',t),\\
\Sigma_{ij,\downarrow}^{TM}(t,t') &= iU^2 T_{ij}(t,t')G_{ji,\uparrow}(t',t),
\end{aligned}
\tag{62}
$$

with the $T$-matrix

$$
T(t,t') = \Phi(t,t') - U \int_{\mathcal{C}} d\bar{t}\, \Phi(t,\bar{t})T(\bar{t},t'),
\tag{63}
$$

where $\Phi_{ij}(t,t') = -iG_{ij,\uparrow}(t,t')G_{ij,\downarrow}(t,t')$. At low densities and strong coupling, this approximation is known to be more accurate [53] than the second Born approximation specified by Eq. (60). In particular, it is expected to mitigate the overly strong damping typical of the second Born approximation in this regime [63]. Expanding the contour time integral as before (cf. Section 2.2.1), one finds Volterra integral equations of the second kind:

$$
\begin{aligned}
T^{\lessgtr}(t,t') &= \Phi^{\lessgtr}(t,t') - U\int_0^t d\bar{t}\,[\Phi^>(t,\bar{t}) - \Phi^<(t,\bar{t})]T^{\lessgtr}(\bar{t},t')\\
&\quad - U\int_0^{t'} d\bar{t}\,\Phi^{\lessgtr}(t,\bar{t})[T^<(\bar{t},t') - T^>(\bar{t},t')].
\end{aligned}
\tag{64}
$$

Solving these alongside Eqs. (16) by the iterative method outlined in Section 3.3, we obtain the results shown in Fig. 12. As expected, the $T$-matrix leads to oscillations that are sustained for far longer than to be concluded from the second Born approximation. We emphasise that these results are computed without further approximations, i.e. without memory truncation or the generalised Kadanoff-Baym ansatz. We find good convergence already at large tolerance values, enabling the computation up to much larger final times than shown here. Particularly when exploring the parameter space of a model, this may serve to save resources.

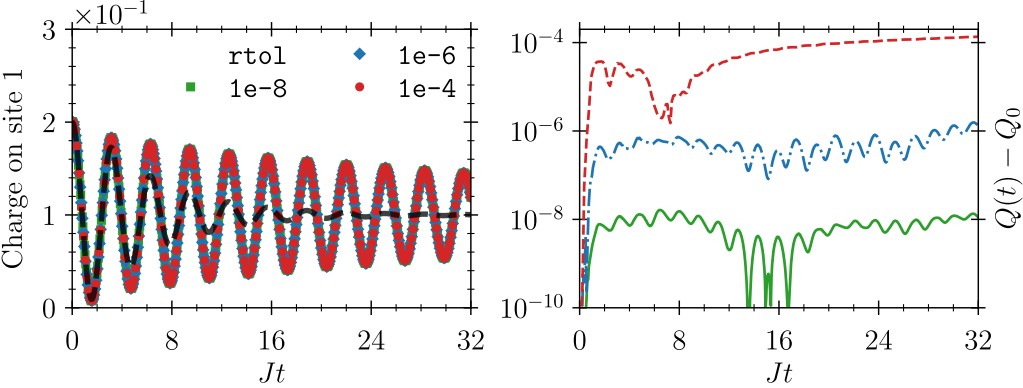

Figure 12: Long-time evolution of the eight-site Fermi-Hubbard lattice (Fig. 8) in the $T$-matrix approximation, with interaction $U/J = 2$ and initial conditions $G_{\uparrow}^<(0,0) = G_{\downarrow}^<(0,0) = i\,\mathrm{diag}(0.1,0.1,0.1,0.1,0.0,0.0,0.0,0.0)$. The initial total charge is hence $Q_0 = 0.8$, while for spin we have $S_0 = 0.0$. Different values of `rtol` are presented, with `atol/rtol=1e-2` in all cases. The (black) dashed line in the left panel shows the second Born approximation. Note that while the results are converged for any of the tolerances chosen here, the conservation laws are adhered to more strictly with more conservative tolerances.

### 4.1.4 Open Bose Dimer

The discussion in 2.2.2 highlighted the close similarity between open Bose systems and the classical Ornstein-Uhlenbeck process. Before transitioning to classical stochastic systems, it is therefore natural to round off the section on quantum dynamics by considering an open quantum system such as the bosonic dimer sketched in Fig. 13. This also has the advantage of requiring the Schwinger-Keldysh formalism in full generality, which is not the case when considering closed quantum systems.

A collection of $L$ bosonic modes $[\hat{a}_i, \hat{a}_i^\dagger] = 1$, $i = 1, \ldots, L$, with on-site energies $\omega_i$ and local Markovian reservoirs at inverse temperatures $\beta_i$, is described by the Lindblad master equation

$$\partial_t \hat{\rho} = -\mathrm{i}\left[\hat{H}\hat{\rho} - \hat{\rho}\hat{H}^\dagger\right] + \lambda \sum_{i=1}^{L}\left[(N_i + 1)\hat{a}_i\hat{\rho}\hat{a}_i^\dagger + N_i\hat{a}_i^\dagger\hat{\rho}\hat{a}_i\right], \tag{65}$$

where $N_i = 1/(e^{\beta_i \omega_i} - 1)$ is the thermal occupation of reservoir $i$, $\lambda$ the system-reservoir coupling, and the (non-Hermitian) operator $\hat{H}$ is defined as

$$\hat{H} = \sum_{i=1}^{L}(\omega_i - \mathrm{i}\lambda(N_i + 1/2))\hat{a}_i^\dagger\hat{a}_i. \tag{66}$$

In the spirit of Section 2.2.2, the Schwinger-Keldysh action equivalent to the master equation (65) reads

$$\begin{aligned} S[\boldsymbol{\phi}(t)] = \sum_{i=1}^{L} \int \mathrm{d}t \; \Big[ &\phi_{i,+}^*(\mathrm{i}\partial_t - \omega_i)\phi_{i,+} - \phi_{i,-}^*(\mathrm{i}\partial_t - \omega_i)\phi_{i,-} \\ &+ \mathrm{i}\lambda(N_i + 1/2)\left(\phi_{i,+}^*\phi_{i,+} + \phi_{i,-}^*\phi_{i,-}\right) \\ &- \mathrm{i}\lambda\left((N_i + 1)\phi_{i,+}\phi_{i,-}^* + N_i\phi_{i,+}^*\phi_{i,-}\right)\Big]. \end{aligned} \tag{67}$$

We assume $\langle\phi\rangle = 0$, which physically corresponds to the absence of a condensate field. For open systems, we need the time-ordered and anti-time-ordered Green functions, which are defined as

$$\begin{aligned} G_{ij}^T(t, t') &= \Theta(t - t')G_{ij}^>(t, t') + \Theta(t' - t)G_{ij}^<(t, t'), \\ G_{ij}^{\tilde{T}}(t, t') &= \Theta(t - t')G_{ij}^<(t, t') + \Theta(t' - t)G_{ij}^>(t, t'). \end{aligned} \tag{68}$$

The equations of motion for the lesser and greater Green functions are now given in compact form by

$$\begin{aligned} \partial_t \boldsymbol{G}^<(t, t') &= -\mathrm{i}\boldsymbol{H}\boldsymbol{G}^<(t, t') + \lambda \operatorname{diag}(N_1, \ldots, N_L)\boldsymbol{G}^{\tilde{T}}(t, t'), \\ \partial_t \boldsymbol{G}^>(t, t') &= -\mathrm{i}\boldsymbol{H}^\dagger\boldsymbol{G}^>(t, t') - \lambda \operatorname{diag}(N_1 + 1, \ldots, N_L + 1)\boldsymbol{G}^T(t, t'), \end{aligned} \tag{69}$$

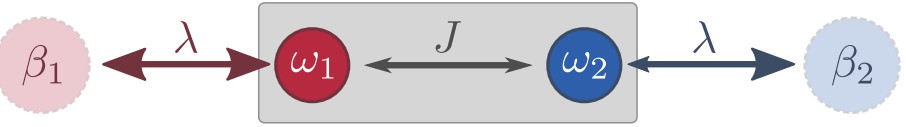

Figure 13: Sketch of a typical non-equilibrium situation with two bosonic modes $\omega_{1,2}$ and reservoirs at inverse temperatures $\beta_1 < \beta_2$.

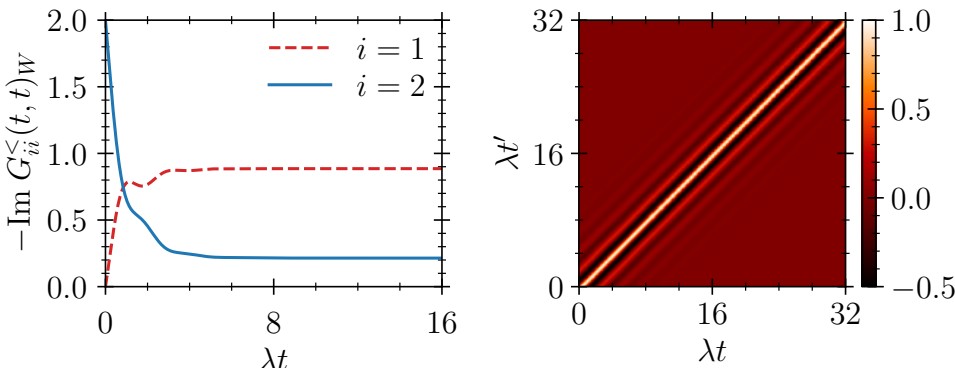

Figure 14: Evolution of the site occupations $\langle \hat{a}_i^\dagger(t)\hat{a}_i(t)\rangle = -\operatorname{Im} G_{ii}^<(t,t)$ and the spectral function $-\operatorname{Im} A_{11}(t,t')$ for the open Bose dimer described by Eq. (65), with $\omega_1 = 2.5\lambda$, $\omega_2 = 0.0$, $J = \lambda\pi/4$, $N_1 = 1.0$ and $N_2 = 0.1$ and initial conditions $G_{ij}^<(0,0) = -2\mathrm{i}\delta_{i2}\delta_{j2}$, $G_{ij}^>(0,0) = -\mathrm{i}\delta_{ij} + G_{ij}^<(0,0)$. The tolerances are `atol=1e-6` and `rtol=1e-4`.

with the non-Hermitian matrix $\boldsymbol{H} = \operatorname{diag}(\omega_1 - \mathrm{i}\lambda(N_1 + 1/2),\ldots,\omega_L - \mathrm{i}\lambda(N_L + 1/2))$. To find the equations of motion on the equal-time diagonal $t = t'$, we have to combine Eqs. (69) with their adjoints, resulting in

$$
\begin{aligned}
\partial_T G_{ij}^<(T,0)_W &= -\mathrm{i}\left[\boldsymbol{H}\boldsymbol{G}^<(T,0)_W - \boldsymbol{G}^<(T,0)_W\boldsymbol{H}^\dagger\right]_{ij} \\
&\quad + \frac{\mathrm{i}\lambda}{2}(N_i + N_j)\left(G_{ij}^<(T,0)_W + G_{ij}^>(T,0)_W\right), \\
\partial_T G_{ij}^>(T,0)_W &= -\mathrm{i}\left[\boldsymbol{H}^\dagger\boldsymbol{G}^>(T,0)_W - \boldsymbol{G}^>(T,0)_W\boldsymbol{H}\right]_{ij} \\
&\quad - \frac{\mathrm{i}\lambda}{2}(N_i + N_j + 2)\left(G_{ij}^<(T,0)_W + G_{ij}^>(T,0)_W\right).
\end{aligned}
\tag{70}
$$

Note also that these equations of motion for the Green functions are agnostic to large occupation numbers (since they are exact). This is not true for approaches based on exact numerics of the density matrix [68], where the Fock space needs to be truncated. For the same reason, a large number of modes $L$ can be handled with a smaller computational effort.

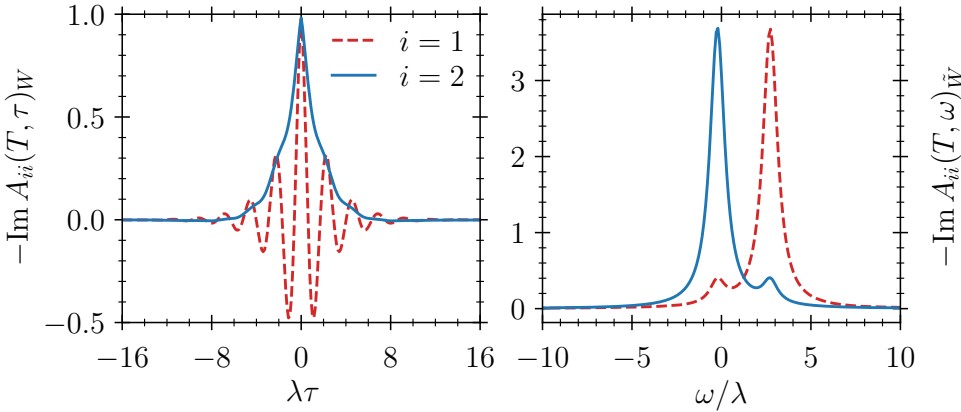

Figure 15: Imaginary parts of the spectral functions $A_{ii}(T,\tau)_W$ and $A_{ii}(T,\omega)_{\tilde{W}}$ at centre-of-mass time $\lambda T = 16$. The system is the open dimer of Fig. 14.

For the sake of this example, as before we now set $L = 2$ and consider the dimer depicted in Fig. 13, where we have also included a hopping term $J(a_1^\dagger a_2 + a_2^\dagger a_1)$. If the two reservoirs are at different inverse temperatures $\beta_1 < \beta_2$, the dimer approaches a steady state in which particles are transported from the hotter to the cooler reservoir via the hopping term. The approach toward the steady state from an initial non-equilibrium distribution is shown in Fig. 14. The spectral function of the "hot" mode $i = 1$, transformed to Wigner coordinates, is also presented in the right panel, highlighting both its stationarity ($T$-independence) and the damped oscillations in the relative-time coordinate $\tau$. Finally, in Fig. 15, a vertical cross section through the right panel of Fig. 14 is taken at $\lambda T = 16$ (i.e. in the centre where a maximal number of data points is available), alongside the corresponding relative-time Fourier transform.

## 4.2 Classical Stochastic Processes

Finally, in this section we turn to two exactly solvable examples from classical stochastic processes, which will provide us with further comparisons of our numerical methods with analytical results. For classical systems, it is convenient to redefine the cumulant-generating function as $W[j, K] = -\ln Z[j, K]$, with the moment-generating function being normalised according to

$$Z = \int \mathcal{D}\boldsymbol{\phi} \exp\{-S[\boldsymbol{\phi}]\}. \tag{71}$$

The corresponding classical 2PI effective action then reads

$$\Gamma[\bar{\boldsymbol{\phi}}, G] = \text{const.} + S[\bar{\boldsymbol{\phi}}] + \frac{1}{2}\text{Tr}\ln G^{-1} + \frac{1}{2}\text{Tr}\,G_0^{-1}[\bar{\boldsymbol{\phi}}]G + \Gamma_2[\bar{\boldsymbol{\phi}}, G]. \tag{72}$$

A natural starting point for an investigation of classical stochastic processes by field-theory methods is, of course, the Gaussian process, for which $\Gamma_2$ vanishes and $G_0^{-1}$ becomes independent of $\bar{\boldsymbol{\phi}}$. We go into some detail on purpose to lower the entry point for applying our algorithm to such problems. Further details on applying the 2PI effective action to stochastic processes can be found in Ref. [79].

### 4.2.1 Ornstein-Uhlenbeck Process

The Ornstein-Uhlenbeck (OU) process [43, 44] is defined by the stochastic differential equation (SDE)

$$dx(t) = -\theta x(t)dt + \sqrt{D}dW(t), \tag{73}$$

where $W(t)$, $t > 0$ is a one-dimensional Brownian motion [44] and $\theta > 0$. The Onsager-Machlup path integral [80]

$$\int \mathcal{D}x \exp\left\{-\frac{1}{2D}\int dt\,(\partial_t x(t) + \theta x(t))^2\right\} \tag{74}$$

is a possible starting point to derive the corresponding MSR action via a Hubbard-Stratonovich transformation. Setting $\boldsymbol{\phi} = (x, \hat{x})^T$, the classical MSR action

$$S[x, \hat{x}] = \int dt\,\left[i\hat{x}(t)(\partial_t x(t) + \theta x(t)) + D\hat{x}^2(t)/2\right], \tag{75}$$

is then equivalent to Eq. (22) after the change of convention introduced in Eq. (71). Note also that we are employing Itô regularisation [28, 46] for simplicity. It is also common to define

a purely imaginary response field $\tilde{x} = i\hat{x}$, which is then integrated along the imaginary axis. Keeping the response field real, the inverse Green function is the Hessian of the action and reads

$$\boldsymbol{G}_0^{-1}(t,t') = \delta(t-t') \begin{pmatrix} 0 & -i\partial_t + i\theta \\ i\partial_t + i\theta & D \end{pmatrix}. \tag{76}$$

The classical saddle-point, exact in this case, is specified via the equations

$$\begin{aligned} 0 &= \delta S[x,\hat{x}]/\delta x|_{x=\langle x(t)\rangle, \hat{x}=\langle \hat{x}(t)\rangle} = -i\partial_t \langle \hat{x}(t)\rangle + i\theta \langle \hat{x}(t)\rangle, \\ 0 &= \delta S[x,\hat{x}]/\delta \hat{x}|_{x=\langle x(t)\rangle, \hat{x}=\langle \hat{x}(t)\rangle} = i\partial_t \langle x(t)\rangle + i\theta \langle x(t)\rangle + D\langle \hat{x}(t)\rangle, \end{aligned} \tag{77}$$

the first of which allows us to set $\langle \hat{x}(t)\rangle = 0$, as required in the MSR formalism [28,46], turning the second equation into the "macroscopic" law of motion [80], i.e. $\partial_t \langle x(t)\rangle = -\theta \langle x(t)\rangle$. In the space spanned by the fields $(x(t), \hat{x}(t))$, the Green function can be written as

$$\boldsymbol{G}_0(t,t') = \begin{pmatrix} F(t,t') & G^R(t,t') \\ G^A(t,t') & 0 \end{pmatrix}, \tag{78}$$

where the retarded Green function $G^R$ and the statistical propagator $F$ are defined as

$$\begin{aligned} G^R(t,t') &= \langle x(t)\hat{x}(t')\rangle, \\ F(t,t') &= \langle x(t)x(t')\rangle - \langle x(t)\rangle\langle x(t')\rangle. \end{aligned} \tag{79}$$

Note that the "variance" $F(t,t')$ is the analogue of the "Keldysh" Green function $G^K(t,t')$ [28,29]. The advanced Green function $G^A$ follows from Eq. (82). The equations of motion of the Green functions are

$$\begin{aligned} \delta(t-t') &= -i\partial_t G^A(t,t') + i\theta G^A(t,t'), \\ \delta(t-t') &= i\partial_t G^R(t,t') + i\theta G^R(t,t'), \end{aligned} \tag{80}$$

admitting the solutions

$$\begin{aligned} G^A(t,t') &= G^A(t-t') = -i\Theta(t'-t)e^{-\theta(t'-t)}, \\ G^R(t,t') &= G^R(t-t') = -i\Theta(t-t')e^{-\theta(t-t')}. \end{aligned} \tag{81}$$

For these classical response functions, there holds the symmetry relation

$$G^A(t,t') = G^R(t',t), \tag{82}$$

which is exploited in the numerical implementation of our algorithm together with the obvious $F(t,t') = F(t',t)$. On the two-time mesh shown in Fig. 3, the equations of motion in the two time directions read

$$\partial_t F(t,t') = -\theta F(t,t') + iDG^A(t,t'), \tag{83a}$$

$$\partial_{t'} F(t,t') = -\theta F(t,t') + iDG^R(t,t'), \tag{83b}$$

respectively, while in Wigner coordinates (s. Appendix A) we find

$$\partial_T F(T,\tau)_W = -2\theta F(T,\tau)_W + iD\left(G^A(T,\tau)_W + G^R(T,\tau)_W\right), \tag{84a}$$

$$\partial_\tau F(T,\tau)_W = \frac{iD}{2}\left(G^A(T,\tau)_W - G^R(T,\tau)_W\right). \tag{84b}$$

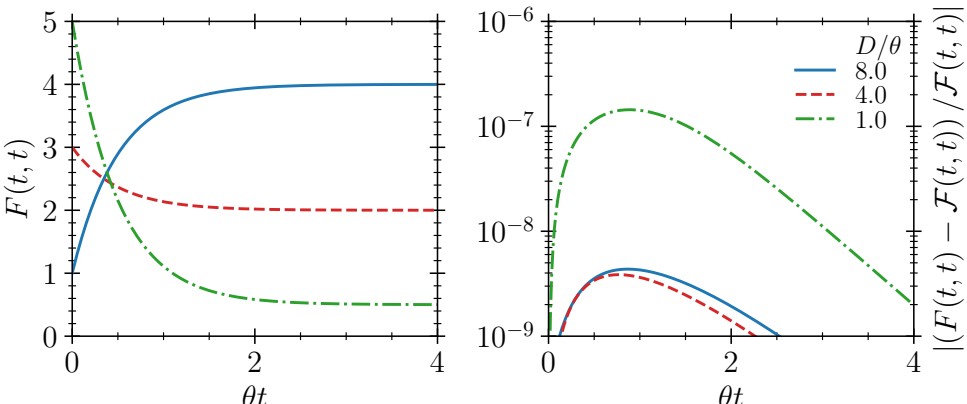

Figure 16: Numerical solution and relative error of Eqs. (85) with `rtol = 1e-7` and `atol = 1e-9`.

To cover the two-time mesh completely, one could in principle use any two of the four Eqs. (83) and (84). Our convention is to pick Eq. (83a) with $t > t'$ for the vertical direction and Eq. (84a) with $\tau = 0$ for the diagonal. Together with the symmetries stated above, the problem is fully determined by the initial conditions and the two equations

$$\partial_t F(t, t') = -\theta F(t, t') + iDG^A(t, t'), \tag{85a}$$

$$\partial_T F(T, 0)_W = -2\theta F(T, 0)_W + D, \tag{85b}$$

where we have used the response identity $G^A(T, 0)_W + G^R(T, 0)_W = -i$, and $G^A(t, t') = 0$ when $t > t'$. For comparison, the analytical solution for the variance or statistical propagator reads

$$\begin{aligned}
\mathcal{F}(t, t') &= \mathcal{F}(0, 0)e^{-\theta(t+t')} - \frac{D}{2\theta}\left(e^{-\theta(t+t')} - e^{-\theta|t-t'|}\right) \\
&= \left(\mathcal{F}(0, 0) - \frac{D}{2\theta}\right)e^{-\theta(t+t')} + \frac{iD}{2\theta}\left(G^A(t, t') + G^R(t, t')\right).
\end{aligned} \tag{86}$$

The numerical results for Eqs. (85) obtained via our adaptive algorithm are shown in Fig. 16 and are in agreement with the analytical solutions.

### 4.2.2 Geometric Brownian Motion

Moving beyond trivial Gaussian systems towards non-linear stochastic processes, a problem of basic interest for financial markets and other systems exhibiting stochastic growth is geometric Brownian motion (GBM). The corresponding SDE is

$$dX(t) = \mu X(t)dt + \sigma X(t)dW(t), \tag{87}$$

where $\mu, \sigma$ are real, and $X(t)$ is usually denoted by $S(t)$ in the financial context. Observe that all formal definitions of the previous Section 4.2.1 carry over with $x(t)$ now replaced by $X(t)$. The analytical solution to Eq. (87) can be found by observing that $\ln X(t)$ is a Brownian motion, i.e.

$$X(t) = X(0)e^{(\mu-\sigma^2/2)t}e^{\sigma(W(t)-W(0))}. \tag{88}$$

The first and second cumulant are thus given by

$$\begin{aligned}
\langle X(t)\rangle &= X(0)e^{\mu t}, \\
\langle X(t)X(t')\rangle &= X(0)^2 e^{\mu(t+t')+\sigma^2 t'},
\end{aligned} \tag{89}$$

where $t \geq t'$. Denoting the response field by $\hat{X}$, the MSR action in Itô regularisation is found to be

$$S[X, \hat{X}] = \int \mathrm{d}t \left[ i\hat{X}(t)(\partial_t X(t) - \mu X(t)) + \sigma^2 X^2(t)\hat{X}^2(t^+)/2 \right]. \tag{90}$$

Since the noise must run "ahead" of the system, we have performed an explicit time-ordering in the non-quadratic part by introducing $t^{\pm} = t \pm \varepsilon$, $\varepsilon > 0$. Expanding around the classical saddle point, we find an inverse Green function

$$
\begin{aligned}
G_0^{-1}(t, t') = {} & \delta(t - t') \begin{pmatrix} \sigma^2 \langle \hat{X}(t) \rangle^2 & -i\partial_t - i\mu \\ i\partial_t - i\mu & \sigma^2 \langle X(t) \rangle^2 \end{pmatrix} \\
& + 2\sigma^2 \langle \hat{X}(t) \rangle \langle X(t) \rangle \begin{pmatrix} 0 & \delta(t^+ - t') \\ \delta(t^- - t') & 0 \end{pmatrix},
\end{aligned} \tag{91}
$$

where we have again explicitly kept track of the necessary time-ordering. Exploiting the diagrammatic properties of the MSR formalism, one can then show that the interacting part of the effective action is exactly given by

$$\Gamma_2 = \frac{\sigma^2}{2} \int \mathrm{d}t \, \langle \delta\hat{X}^2(t) \rangle F(t, t), \tag{92}$$

where $\delta\hat{X}$ denotes the fluctuations of the response field, and all other diagrams indeed vanishing identically. Consequently, the first cumulant (the "mean-field") evolves according to

$$
\begin{aligned}
0 = \frac{\delta\Gamma}{\delta\langle\hat{X}(t)\rangle} = {} & (i\partial_t - i\mu)\langle X(t)\rangle + \sigma^2 \langle\hat{X}(t)\rangle\langle X(t)\rangle^2 + \sigma^2 \langle X(t)\rangle \left[ G^A(t^+, t) + G^R(t^-, t) \right] \\
= {} & (i\partial_t - i\mu)\langle X(t)\rangle,
\end{aligned}
$$

where the time-ordering ensures that the response functions do not contribute. The statistical propagator obeys the two equations

$$
\begin{aligned}
i\partial_t F(t, t') = {} & i\mu F(t, t') - \sigma^2 \left[ \langle X(t)\rangle^2 + F(t, t) \right] G^A(t, t'), \\
-i\partial_{t'} F(t, t') = {} & -i\mu F(t, t') + \sigma^2 \left[ \langle X(t')\rangle^2 + F(t', t') \right] G^R(t, t'),
\end{aligned} \tag{93}
$$

which result in the $t = t'$ equation

$$\partial_T F(T, 0)_W = 2\mu F(T, 0)_W + \sigma^2 \left[ \langle X(T)\rangle^2 + F(T, 0)_W \right]. \tag{94}$$

Exemplary results for a number of different noise strengths $\sigma$ and initial conditions are presented in Fig. 17 and are in agreement with the analytical solutions from Eq. (89).

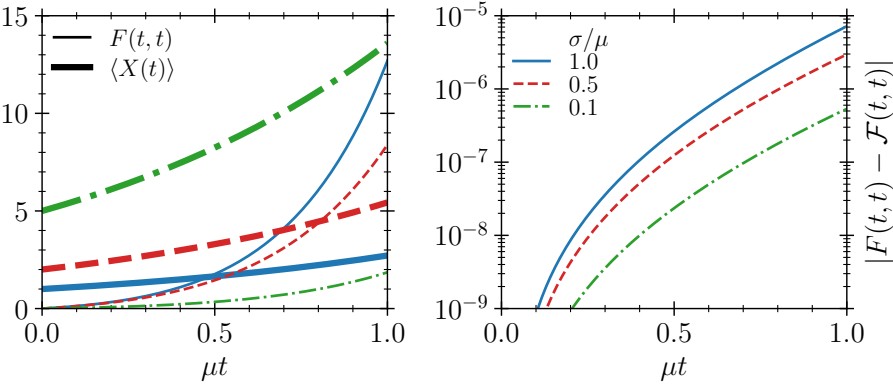

Figure 17: Numerical solution and absolute error of Eqs. (93) with `rtol = 1e-7` and `atol = 1e-9`, where $\mathcal{F}(t, t)$ again denotes the analytical solution for the second cumulant.

## 5 Conclusion

We have presented a numerical algorithm implementing adaptivity in the step size and the integration order for the solution of (quantum) field theories described by Kadanoff-Baym equations with their typical two-time dependence (Section 3). We have performed an error analysis on an exactly solvable quantum system. It confirms that the algorithm can determine an optimal step size and, thus, significantly reduces the number of time-steps while also minimising the error (cf. Fig. 5) in comparison with fixed-step methods. As a field-theory approach, our method is universally applicable to both quantum and classical systems. We have demonstrated this for small but interacting quantum models, namely an interacting Bose gas (Section 4.1.2) and a fermionic Hubbard system (Section 4.1.3), as well as for an open quantum system (Section 4.1.4) and classical stochastic processes (Section 4.2). Detailed work on systems with very long memory times, such as Kondo systems, is forthcoming.

An intricate yet promising research avenue appears to be the solution of the integro-differential equations in Wigner coordinates together with the implementation of independent grids in the centre-of-mass and relative times, respectively. As we observed, the time scales in these two directions can strongly differ. Providing adaptivity in this respect could possibly improve on existing truncation schemes.

Incorporating higher-order cumulants (i.e. vertex corrections) self-consistently into time-stepping schemes as presented here is a research problem on which little progress seems to have been made. The corresponding equations can, however, be derived via known techniques such as the four-particle irreducible effective action [81]. The mounting numerical cost could potentially be compensated via memory truncation, in particular for systems with large relative-time relaxation rate.

It remains to be seen whether the present approach via explicit equations of motion for the time-dependent cumulants can provide added value when solving non-linear classical stochastic processes, as opposed to more conventional techniques based on the discretisation of stochastic differential equations. Since our implementation readily supports classical Green functions, it can serve as a basis for further investigation into this direction.

An open-source implementation of our algorithm in the scientific-computing language Julia [82] is available at https://github.com/NonequilibriumDynamics/KadanoffBaym.jl.

## Acknowledgements

**Funding information** We acknowledge funding from the Deutsche Forschungsgemeinschaft (DFG) within the Cooperative Research Center SFB/TR 185 (277625399) and the Cluster of Excellence ML4Q (390534769).

## A Wigner Coordinates

The time-dependent spectral function is defined as

$$A(t, t') = \left[ G^>(t, t') - G^<(t, t') \right]. \tag{95}$$

By a rotating and squeezing the original time coordinates the Wigner coordinates are obtained

$$\begin{pmatrix} \sqrt{2} & 0 \\ 0 & \frac{1}{\sqrt{2}} \end{pmatrix} \cdot \begin{pmatrix} \cos\frac{\pi}{4} & -\sin\frac{\pi}{4} \\ \sin\frac{\pi}{4} & \cos\frac{\pi}{4} \end{pmatrix} \cdot \begin{pmatrix} t \\ t' \end{pmatrix} = \begin{pmatrix} 1 & -1 \\ \frac{1}{2} & \frac{1}{2} \end{pmatrix} \cdot \begin{pmatrix} t \\ t' \end{pmatrix} =: \begin{pmatrix} \tau \\ T \end{pmatrix}, \tag{96}$$

and the Wigner transform is defined as

$$A(T, \tau)_W = A(T + \tau/2, T - \tau/2) \,. \tag{97}$$

Generally $T$ is called the *centre-of-mass* time and $\tau$ the *relative* time. Their derivatives are given by

$$\partial_T = \partial_t + \partial_{t'}, \quad \partial_\tau = \frac{\partial_t - \partial_{t'}}{2} \,. \tag{98}$$

Despite being a simple operation, considering two-time observables in Wigner coordinates is worthwhile due to their intrinsic physical significance. For example, a time-translation invariant problem (such as an equilibrium problem) is independent of $T$.

Furthermore, the Wigner-Ville transform

$$A(T, \omega)_{\tilde{W}} = \int d\tau e^{i\omega\tau} A(T, \tau)_W = \int d\tau e^{i\omega\tau} \left[ \int \frac{d\omega'}{2\pi} e^{-i\omega'\tau} A(T, \omega')_{\tilde{W}} \right] \tag{99}$$

is the generalisation of the equilibrium spectral function and can roughly describe how the spectral density changes with the centre-of-mass time $T$.

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
