# Peer review of "Adaptive Numerical Solution of Kadanoff-Baym Equations"

_SciPost Physics, doi:SciPost Phys. Core 5, 030 (2022)_

## Round 1 · Referee Report · Anonymous (Referee 1) · 2021-11-26

Strengths

-- Clear presentation, including a concise summary of the theory background.
-- Potentially useful technical developments for approaching many-body dynamics of quantum and classical problems by an adaptive choice of time steps.

Weaknesses

-- Worked examples not clearly demonstrating the advantage of the new method (so the potential use is not fully backed/proven, from the examples).
-- Seems to be limited to special classes of models, where the interaction can be treated perturbatively (in the spirit of self-consistent Born, no perspective on inclusion of vertex corrections given).

Report

I would recommend publication of the paper after the following points of criticism have been addressed:

Introduction: Why should it be enough to study the two-time correlator? This obviously will require approximations truncating the full Dyson-Schwinger hierarchy. Some comments/anticipation would be useful, already in the introduction.

Reduction Schwinger-Keldysh --> MSR: The example of a free theory is correctly worked out, but a word of caution should be placed in what concerns this reduction in the case of interacting problems: There, it can only be achieved in the frame of further approximations, neglecting terms with 'quantum' fields entering at order higher than two.

3.3.2, memory truncation: should be re-emphasized that the Green's function are connected, therefore one has clustering properties and decay as a function of the relative time coordinate. Interesting would be a discussion of the difference between algebraic (expected in the presence of conservation laws and / or quantum problems) and exponential decay in that coordinate, how does this affect the scaling of the method?

Choice of example models: What is the reason to choose the excitation transfer model? It looks to me that choosing a more generlic (e.g. Bose-Hubbard) model would likewise be suitable to do the benchmark calculations? Later it becomes clear (avoiding hopping in the single-particle sector), but the logic of choice of models should be anticipated.

In the same vein, the paper would benefit very much from more impressive examples. The breadth of examples is an asset, but they are all constrained to non-interacting systems or interacting ones on very few sites (often times, 2). This does not make a convincing case for a new method. Is it not possible to consider, say, and interacting scalar field theory in spatial continuum, see e.g. the cited works by the Berges group, and compare the new technique to previous ones?
The applicability of the method to generic situations (i.e. beyond perturbation theory in the interactions) is not clear, and not even an effort is made to comment on that. This is key in non-trivial out-of-equilibrium dynamics, such as the description of non-thermal fixed points. Some comparison to the so far successfully developed techniques to tackle such problems would be useful.
An answer could be that the latter rely on an effectively classical description due to high mode occupations. Could the present technique also benefit from that?

A more precise / extensive discussion of the advantage obtained from the adaptive method would be helpful, and a service to practitioners who might want to use the method. The only statement I could find about that comparison is in the first example but is very short and qualitative. Are there no scaling arguments available?

Requested changes

see above

---

## Round 1 · Referee Report · Anonymous (Referee 2) · 2022-1-29

Strengths

1- Good overview over non-equilibrium field theory and the state-of-the art numerical methods used 2- Connection between classical and quantum, closed and open system dynamics emphasized 3- Innovative implementation using adaptive numerical algorithm 4- Code publicly available in next-gen high-performance language julia 5- Implementation of true non-equilibrium setting of KBE, opposed to the previously available c++ package NESSI, which focusses on thermal initial conditions.

Weaknesses

1- No new physics results, advance is purely numerical 2- Github repo not publicly available (yet?) 3- Only small systems studied 4- Only perturbative approximations, no memory integrals in self-energy evaluation considered

Report

First of all, I would like to apologize to the authors for the late submission of this report.

In this paper, the authors present a new algorithm to integrate two-time integro-differential equations. They apply their algorithm to the two-time Kadanoff-Baym equations encountered in treating dynamics of many-body systems in classical and quantum, closed and open systems. They introduce the formalism of nonequilibrium quantum field theory in a unified manner for both Bosons and Fermions and show its connections to the treatment of open systems as well as classical stochastic dynamics. They benchmark their method on several systems, notably free Fermions dynamics, the Fermi-Hubbard model within the self-consistent Born approximation, mixtures of Bosons, a bosonic open system as well as classical stochastic time evolution.

I personally very much liked the presentation of the authors, who tried to clean up the at times confusing notations used in the field. Their algorithm could lead to an application of Kadanoff-Baym equations to previously inaccessible regimes and therefore deserves a presentation to the community.

However, this is not done in the paper in its present form, i.e. no new physical results are presented. Using adaptive step sizes, while technically challenging to implement, is by no means a new development in the integration of differential equations. To me, the paper serves an important purpose as a reasonably self-contained introduction to Kadanoff-Baym equation methods that could be used for example for new students to enter the field. However, due to the unavailability of the advertised GitHub repository and the lack of a deeper documentation of the alleged repository, the paper does not serve as a self-contained introduction that can be used before using the advertised code. This paper, alongside the code, could replicate the immense success of the introduction to TeNPy (https://scipost.org/10.21468/SciPostPhysLectNotes.5), as a intro+code ressource used by students and researchers to quickly obtain KBE results and create a similar community that can work on maintaining and extending the code.

In summary, I believe that there is too little physics advancement to meet the requirements of SciPost Physics. However, if the paper was amended with a (short) overview of the julia code (which should be made available prior to publication), I think it would be perfectly suited for SciPost Physics Codebases or LectureNotes.

Requested changes

1- Unify the basis for the Green's functions (GF). In the general section, G^<, G^> are used, but then at different stages, G^R, G^A, K, F, G^T, G^{\tilde T}, ... are introduced. This is as far as I can see unnecessary and confusing. It would improve the readability of the paper immensily, if only two linearly independent GF were used (my preference is the F/Rho notation used by Berges in Ref.[2], but this is of course a question of taste)

2- When discussing the fact that GF decay as a function of relative time, the GKBA is often referenced as an interpretation. To me, that's just another numerical method, not a physical interpretation. The reason that GFs decay as a function of rel. time is the fact that closed, interacting quantum systems thermalize, which implies a loss of memory of the initial state. See e.g. Berges, Cox, Physics Letters B 2001. This could be emphasized.

3-I'm not sure I understand why in section 3.3.2. it is said that the numerical effort with a memory cut scales as n^2. The point of the memory cut is that the numerical effort of the RHS of the KBE is constant, and the t,t' plane in which the Green's function is evualated has the fixed dimension N_tau^2 which means that in the presence of a memory cut, KBE always scales linearly with n. As a memory cut can be employed in almost all physical applications due to the presence of thermalization (see point 2), I find it a bit surprising to say that "KBE integration scales at least like n" in the intro and abstract. I would say at least like n and in special cases like n^2 or n^3.

4- In the intro, 1/N expansions are mentioned. They require to solve an integral equation to evaluate the self-energy, which increases the numerical effort by a factor of n or N_tau. As a lot of interesting physical applications require such approximations (as they are valid in non-perturbative settings), it would be great to at least comment if not explicitly calculate how the adaptive algorithm can be generalized to this case.

5- The excellent review of Schlünzen et al., doi:10.1088/1361-648X/ab2d32, should be cited rather prominently as it gives a much more in-depth and complete introduction into the same methods.

6- The efficiency of the implementation also depends on the scaling with system size L. It would be great to show that it is also optimal, i.e. the various matrix/tensor contractions were implemented well.

7- The Github repo should be made public before publication and a short documentation of the structure of the package should be added.

8- A central point of the paper is the connection between quantum time evolution and classical stochastic evolution. There is however another limit, namely the classical limit of closed quantum systems, as e.g. reviewed in Ref. [2]. It would be great if the authors could comment on this limit, in particular how the self energies could be altered to yield an approximation to the classical field theory.

9- I'm not sure Eq.(6,10) is consistent with Eq.(1). Shouldn't there be a factor of 2 if the Grassmann Fermionic fields are decomposed into two real Majoranas? Eq.(6) is as far as I can see only valid for complex Grassmann fields, which is at odds with the claim that everything in this section is decomposed into real and imaginary parts. See e.g. section 2.2.1 in the Thesis of Schuckert, https://mediatum.ub.tum.de/1621753. Using the Majorana notation, the presentation could be compressed by simply writing \pm for real Bosons/Fermions in 5,6 and 9,10.

---

## Round 1 · Referee Report · Anonymous (Referee 3) · 2022-1-30

Strengths

1)New numerical method for the solution of Kadanoff-Baym Equations. Adaptive time stepping leads to a reduction of computational costs/increase of maximum simulation time
2)General method, can be applied to a variety of problems in classical/quantum many body systems
3)Introduction to the formalism and several applications

Weaknesses

1)Section 3, the core part of the manuscript, could be expanded and clarified in notation. In particular section 3.2. Not easy to understand how to choose the time-step in the adaptive scheme.
2)Comparison with other approaches could be improved.

Report

The manuscript by Meirinhos et al focuses on the nonequilibrium dynamics of classical and quantum systems and presents a new numerical approach for the solution of Kadanoff-Baym (KB) equations. These are the basic equations governing the dynamics of nonequilibrium Green's functions and as such emerge in a wide variety of settings.

The authors present first an introduction to the formalism of nonequilibrium field theory, needed to derive the structure of the KB equations. In Section 3 they describe the numerical method used to solve the equations and in Section 4 present their results. In particular the authors focus on different applications taken from both the classical stochastic field theory context and from the physics of closed and open quantum systems. The main result of this work is that their adaptive time-step method allows to reduce the computational cost of dynamical simulations, thus allowing for longer time scales to be reached.

In my opinion this is a nice work, which addresses an important problem of wide applicability and provides an advance towards its efficient solution. I appreciate in particular the discussion on the theoretical background and the variety of examples considered by the authors. I have few requested changes, mainly concerning the presentation of the results.

Requested changes

1)In the balance of the work I find that Section 3 could be expanded, in particular the core sections where the method is described. At the moment they seem a bit hard to read. In particular it is not obvious how in practice the method work and how the time step is chosen. I suggest the authors to expand and clarify the discussion around Eq.39. A more synthetic/less verbose description of the algorithm in practice, could be helpful for the reader

2)Section 4 contains several examples and applications, which is certainly a nice aspect of the work. However I think the authors could improve this part of the manuscript for what concerns the advantage that their method has over conventional approaches.

Figure 5 and Table I are good, although: the table itself is rather hard to read without an appropriate caption. Various symbols used there have been introduced in previous sections and it is hard to keep up for the reader. Furthermore, one wonders why a similar analysis is not performed in the other cases. Overall the question of whether this approach should be preferred to more conventional ones need to be addressed more directly.

---

## Round 2 · Referee Report · Anonymous (Referee 1) · 2022-4-5

Report

I have gone through the reply of the authors and the amendments to the text. The authors have satisfactorily replied to my comments and criticisms and I thus recommend publication of this work.

---

## Round 2 · Referee Report · Anonymous (Referee 2) · 2022-4-6

Report

The authors have substantially improved the paper in the aspects I had commented on. The code has now been published. However, in its current form it will be of little use to the community because it is not enough documented, in particular the examples are not usable in its current form. It would probably increase the reach of this code, the number of people using it and therefore citing this paper, if the authors improved on the documentation.

The central novelty of this paper is the application of adaptive time stepping to KBE, which is a purely algorithmic/numerical progress and no new results on physical phenomena is presented. Therefore, I would still recommend rather publishing this paper in either Codebase or Core.

---

## Round 2 · Referee Report · Anonymous (Referee 3) · 2022-4-19

Report

Dear Editor,
I have read the Authors reply to my previous report and the new manuscript. I think the Authors have successfully addressed the points I previously raised and improved their manuscript accordingly. I think this work is now suitable for publication. Given the acceptance criteria of Scipost I think Scipost Physics Core would be more appropriate in this case.

---

## Round 2 · Author Response

We thank the referees for reviewing our manuscript in detail and for their constructive remarks. In this resubmission, we address the comments and requested changes, and accordingly list all major modifications to the original manuscript. These will be posted as a comment for better visualisation.

---

## Round 2 · List of Changes

See "Author comments".

---

## Editorial Decision

published